



# Objective identification of high-wind features within extratropical cyclones using a probabilistic random forest (RAMEFI). Part I: Method and illustrative case studies

Lea Eisenstein[1], Benedikt Schulz[2], Ghulam A. Qadir[3], Joaquim G. Pinto[1], and Peter Knippertz[1]

[1]Institute of Meteorology and Climate Research, Karlsruhe Institute of Technology, Karlsruhe, Germany
[2]Institute for Stochastics, Karlsruhe Institute of Technology, Karlsruhe, Germany
[3]Heidelberg Institute of Theoretical Studies, Heidelberg, Germany

**Correspondence:** Lea Eisenstein (lea.eisenstein@kit.edu)

**Abstract.** Strong winds associated with extratropical cyclones are one of the most dangerous natural hazards in Europe. These high winds are mostly associated with five mesoscale dynamical features, namely the warm (conveyor belt) jet (WJ), the cold (conveyor belt) jet (CJ), cold-frontal convective gusts (CFC), strong cold sector winds (CS) and - at least in some storms - the sting jet (SJ). The timing within the cyclone's lifecycle, the location relative to the cyclone core and some further

characteristics differ between these features and hence likely also the associated forecast errors. Here we present a novel objective identification approach for these high-wind features using a probabilistic random forest (RF) based on each feature's most important characteristics in near-surface wind, rainfall, pressure and temperature evolution. As CJ and SJ are difficult to distinguish in near-surface observations alone, these two features are considered together here. A strength of the identification method is that it works flexibly and independent of spatial dependencies and gradients, such that it can be applied to irregularly

spaced surface observations and to gridded analyses and forecasts of different resolution in a consistent way. As a reference for the RF, we subjectively identify the four storm features (WJ, CS, CFC, CJ+SJ) in 12 winter storm cases between 2015 and 2020 in both hourly surface observations and high-resolution reanalyses of the German COSMO model over Europe, using an interactive data analysis and visualisation tool. The RF is then trained on station observations only. The RF learns physically consistent relations and reveals mean sea-level pressure (tendency), potential temperature, precipitation amount

and wind direction to be most important for the distinction between the features. From the RF we get probabilities of each feature occurring at the single stations, which can be interpolated into areal information using Kriging. The results show a reliable identification for all features, especially for WJ and CFC. We find difficulties in the distinction of CJ and CS in extreme cases, as the features have rather similar meteorological characteristics. Mostly consistent results in observations and reanalysis data suggest that the novel approach can be applied to other data sets without a need of adaptation. Our new method

RAMEFI (RAndom-forest based MEsoscale wind-Feature Identification) is made publicly available for straightforward use by the atmospheric community, and enables a wide range of applications, e. g., towards a climatology of these features for multi-decadal time periods (see Part II), analysing forecast errors in high-resolution COSMO ensemble forecasts and to develop feature dependent postprocessing procedures.





## 1 Introduction

In the mid-latitudes, extratropical cyclones belong to the most severe natural hazards, especially during wintertime. These winterstorms can cause high wind speeds, heavy precipitation, storm surges and thus considerable damage. Prominent examples for Central Europe are *Lothar* (December 1999; Wernli et al., 2002) and *Kyrill* (January 2007; Fink et al., 2009). The development of extratropical cyclones is usually portrayed on the basis of the Norwegian (NC; Bjerknes, 1919) or the Shapiro-Keyser cyclone model (SKC; Shapiro and Keyser, 1990). Both types of cyclones evolve along a frontal wave and include the formation

of a warm front followed by a cold front with a warm sector in between. While in NCs the cold front slowly catches up with the warm front resulting in an occluded front, SKCs develop a frontal fracture (as illustrated in Fig. 1) and ultimately a warm seclusion when the warm front, then often referred to as a bent-back front, wraps around the cyclone centre.

High winds are typically associated with four mesoscale features within the synoptic-scale cyclone (Fig. 1): the warm conveyor belt jet or short warm jet (WJ), the cold conveyor belt jet, short cold jet (CJ), cold-frontal convection (CFC) and the

sting jet (SJ), which only occurs within SKC. Furthermore, high wind speeds are often detected within the cold sector (CS) without the formation of a distinct mesoscale feature. As the cold front itself, the cold sector is usually convectively active (labelled pCFC in Fig. 1), leading to downward momentum transport in the vicinity of showers or even thunderstorms.

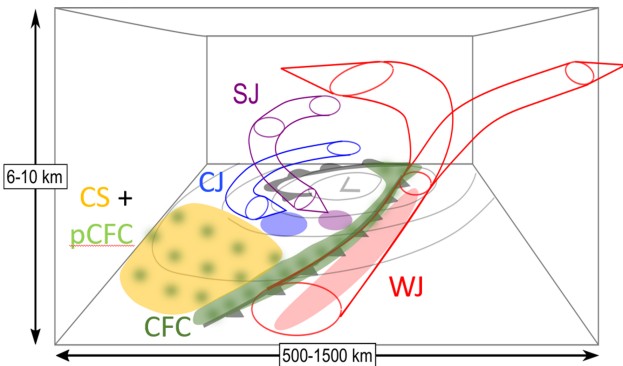

**Figure 1.** Conceptual model of the 3D structure of a SKC showing the WJ (red), CJ (blue) and SJ (magenta). In each case, the region of strong surface winds is indicated by an ellipse. Figure and caption adapted from Clark and Gray (2018) (their Fig. 7) to include CFC (green) and CS (gold) with embedded post-cold-frontal convection (pCFC; light green).

All features can cause damage due to strong gusts, such that it is important to accurately forecast them and their associated windfields. A widely employed approach to improve forecasts (not only of wind) is statistical post-processing (Vannitsem et al.,

2021), where the model output is corrected on the basis of past forecast errors. The performance of various post-processing methods for wind gusts is discussed in a recent paper by Schulz and Lerch (2022), who find that novel neural network-based approaches significantly improve forecast reliability and accuracy. However, Pantillon et al. (2018) analysed the post-processing of wind gusts of several winter storms over Germany and found that in some cases post-processing can actually considerably worsen the forecast, as it did, e. g., for storm *Christian* (October 2013), which is known to have developed an SJ (Browning





et al., 2015). Similar results were also found for storm *Friederike* in January 2018 (not shown). This can generally come from a misprediction of the cyclone track or intensity but could also indicate that the characteristics of individual mesoscale high-wind features are not well represented. If that was the case, a feature-dependent post-processing could lead to further improvement, as it could take into account the specific dynamical characteristics and how they are treated in the forecast model. By developing an objective identification algorithm for the wind features shown in Fig. 1, this work lies the foundation for further exploring this idea.

Hewson and Neu (2015) analysed observations and reanalysis data of 29 wind storms with a focus on the three low-level jets, that is, WJ, CJ and SJ. They included CFC in their WJ analysis instead of treating it as an independent feature. They showed that the three wind features differ in their location of occurrence relative to the cyclone centre, in their timing within the cyclone lifecycle, as well as in their duration, strength, surface footprint and more. Furthermore, Earl et al. (2017) looked at the most common causes of high surface gusts in UK extratropical cyclones and, besides WJ, CJ and SJ, included several convection-induced high-wind features. They based the identification of SJs on satellite images, the location of the gusts within the cyclone and the deepening rate. Since no confirmation with Lagrangian trajectory analysis was done, the identified features are referred to as "potential SJs". Earl et al. (2017) found that although WJs and CJs are the most common causes of high winds, the strongest gusts are caused by CFC and potential SJs. Parton et al. (2010) categorised strong wind events captured by a wind-profiling radar in Wales over a 7-year period into cold frontal events (similar to our definition of CFC), warm sector winds (similar to WJ), tropopause folds/warm fronts, SJs and unclassified events. They found that warm sector events were the cause of around $40\,\%$ of all strong winds followed by cold frontal events with around $24\,\%$.

As all these approaches are purely subjective and relatively time consuming and thus hard to automate, we aim to develop an objective analysis of the different mesoscale wind features that can flexibly be applied to station and gridded data and thus serve as a basis for climatological studies, forecast evaluation and postprocessing development. The strategy we follow is to start with a subjective identification (as in previous studies) but to use the results to then train a probabilistic random forest (RF) to develop an objective procedure that can be applied to cases outside of the training data set. The identification is designed to be independent of horizontal gradients, hence resolution, and can principally be applied to observations from a single weather station. In addition, the identification is based on tendencies over $1\,\mathrm{h}$ only, making it applicable to timeseries with gaps. Given that the provision of such a feature-dependent post-processing tool can enhance the forecasts of strong winds and wind gusts, it can potentially contribute towards better weather warnings and impact forecasting of such events (e. g., Merz et al., 2020). In this paper we will show examples using surface stations and COSMO reanalysis data. A full long-term climatology is the focus of Part II of this paper. The output of the RF are feature probabilities rather than binary identification, which allows an evaluation of how good individual data points fit the typical feature characteristics and the identification of hybrid features or transition zones.

The paper is structured as follows: First, a short description of each wind feature is given in Sect. 2 based on results from previous studies, before the used data sets are discussed in Sect. 3. Section 4 then details the used methods, starting from the subjective labelling of wind features through the training of the RF to the display of areal feature information. While Sect. 5 illustrates our approach with a case study, a statistical evaluation of the performance of the RF can be found in Sect. 6.



Strengths and weaknesses of the approach are discussed on the basis of specific case studies in Sect. 7. In Sect. 8 we finally summarise our work and discuss future plans.

## 2  Characteristics of high-wind features

In this section the most important high-wind features within extratropical cyclones (Fig. 1) are shortly described on the basis of the existing scientific literature. While WJs and CJs are common to both cyclone models, SJs can only occur in SKC (Clark

and Gray, 2018). On the other hand, SKC are rarely accompanied by (strong) CFC, since their cold front is often weak (Schultz et al., 1998). The description is mostly based on Hewson and Neu (2015), Earl et al. (2017) and Clark and Gray (2018). For more details, the reader is referred to these publications.

### 2.1  Warm (conveyor belt) jet

As the name implies, the WJ is associated with a warm air flow, typically ahead of and later ascending above the surface

cold front, often referred to as the warm conveyor belt (Fig. 1; Wernli and Davies, 1997; Eckhardt et al., 2004; Madonna et al., 2014). Located in the warm sector of the cyclone, the WJ is usually characterised by positive temperature anomalies, decreasing pressure with time and little or no precipitation. The WJ is usually associate with the first strong winds starting in an early stage of a cyclone. Maximum gusts of around $25\,\mathrm{m\,s^{-1}}$ are typical (Hewson and Neu, 2015). Since the warm conveyor belt is an ascending airstream, the winds at the surface weaken and disappear in later stages when the warm conveyor belt

no longer affects the boundary layer. The jet is long-lived with a duration of $24\,\mathrm{h}$ to $48\,\mathrm{h}$ and can cause a large surface wind footprint with a width of $200\,\mathrm{km}$ to $500\,\mathrm{km}$ and length of up to $1000\,\mathrm{km}$ (Hewson and Neu, 2015). While the predictability was evaluated to be good with a relatively high coherence in space and time, Hewson and Neu (2015) found that the occurrence of very high winds within the warm sector are rather unusual, while Parton et al. (2010) associates $40\,\%$ of strong wind events with winds within the warm sector. Such a seeming contradiction can potentially be resolved with a climatological application

of the identification algorithm we develop in this paper. Due to generally stable conditions in the warm sector, wind speeds above the boundary layer are usually much higher than surface gust speeds. Compared to the CJ and SJ, the WJ is the most long-lasting, but – as already mentioned – typically does not cause the most destructive winds.

### 2.2  Cold (conveyor belt) jet

The CJ is associated with the main airflow of the cold conveyor belt that turns cyclonically around the centre of the low. At

first, the CJ moves around the northwestern flank behind the occluded or bent-back front beneath the cloud head. Since it is travelling against the motion of the low-pressure system, the CJ is hard to see in Earth-relative winds until it wraps around the cyclone centre. This usually happens around the time the maximum intensity is reached. The CJ weakens when the low decays or shortly before that. The jet mainly stays close to the ground $850\,\mathrm{hPa}$ (Smart and Browning, 2014) or is ascending slightly during its lifecycle (Martínez-Alvarado et al., 2014). With typical maximum gusts around $30\,\mathrm{m\,s^{-1}}$ the CJ is stronger than the

WJ but with a typical lifetime of $12\,\mathrm{h}$ to $36\,\mathrm{h}$ does not last as long (Hewson and Neu, 2015). The impacted area expands with





time, while the cold conveyor belt, and hence the CJ, wraps around the cyclone centre when finally the footprint can reach a width of around $100\,\mathrm{km}$ to $800\,\mathrm{km}$ and a length of up to $2500\,\mathrm{km}$ (Hewson and Neu, 2015). Although forming later in the lifecycle of the parent cyclone than the SJ, both jets can coexist. A damaging CJ is more common than an SJ or WJ over Europe (Hewson and Neu, 2015).

### 2.3 Sting jet

The SJ is a distinct airstream that descends from mid-levels within the cloud head into the frontal fracture region of an SKC (Fig. 1). When the CJ wraps around the low, the SJ can be replaced by the CJ or merge with it. A detailed review can be found in Clark and Gray (2018). The most common way to identify SJs is to compute Lagrangian trajectories (e. g., Volonté et al., 2018; Eisenstein et al., 2020), for which a high resolution is required, horizontally and vertically, but also in time. Recent work has tried to identify SJs with low-cost approaches. For example, Gray et al. (2021) introduced an instability-based precursor tool, while Manning et al. (2022) developed a kinematic approach looking for reversals in the vertical gradient of horizontal wind speed along streamlines. The distinction between the SJ and CJ in surface observations alone is challenging as we will discuss in Sect. 4.1.

### 2.4 Cold-frontal Convection

The passage of cold fronts is often accompanied by heavy precipitation, sometimes in form of convective lines, which, in turn, can cause strong wind gusts associated with the downward transport of high momentum from above the boundary layer. Cold frontal rain, snow and graupel is responsible for around $28\,\%$ of extreme precipitation events in the midlatitudes (Catto and Pfahl, 2013). The frontal zone is characterised by a marked change in wind direction and a decrease in temperature (Clark, 2013). In extreme cases, tornadoes can occur in association with CFC, causing even more hazardous winds (e. g., *Kyrill*; Fink et al., 2009). Earl et al. (2017) suggest a separation of CFC into convective lines and pseudo-convective lines. The latter do not strictly satisfy identification criteria for convective lines but show characteristics of organised, strong convection and may fulfil the criteria earlier or later. A convective line shows a clear signal in radar imagery at $3\,\mathrm{km}$ to $10\,\mathrm{km}$ height (Parton et al., 2010). In case of a kata cold front, CFC occurs at and ahead of the surface cold front (Lackmann, 2011). Although high winds might then strictly speaking occur within the warm sector, we have decided not to associate it with the WJ due to the physically distinct characteristics. In case of an ana cold front, CFC occurs at and behind the cold front, and the distinction is therefore more straightforward. SKC usually have a rather weak cold front, hence high winds are less likely to be associated with CFC (Schultz et al., 1998).

### 2.5 Cold sector winds and post-cold-frontal convection

The cold sector is the region behind the cold front. In this area high winds can be caused by post cold-frontal convection and also by a so-called dry intrusion. This region of a cyclone is generally being known for its instability and turbulent behaviour. A dry intrusion is, as an SJ, a descending airstream but in this case one that originates near the tropopause or even in the





lower stratosphere. This way it brings dry air down to the middle and lower troposphere moving towards the cyclone centre (Raveh-Rubin, 2017). Dry intrusions – and later in the cyclone's lifecycle a dry slot near the cyclone centre – can often be seen in water vapour satellite imagery. Raveh-Rubin (2017) state that dry intrusions can cause destabilisation and increased wind

gusts. Furthermore, cold fronts show a stronger temperature gradient, higher winds and more precipitation when accompanied by a dry intrusion (Catto and Raveh-Rubin, 2019; Raveh-Rubin and Catto, 2019). Most publications consider all high-winds on the colder side of a cyclone to be a CJ or SJ and do not distinguish these features to other cold-sector winds (e. g., Manning et al., 2022). Given the relatively large area of the cold sector in many cyclones and the appearance of discernible substructures, here we decided to specifically separate out CFC, CJ and SJ, and then label the remaining strong winds as CS.

## 3 Data

### 3.1 Surface observations

The main basis of our analysis is a data set of hourly surface observations from 2001 to mid-2020. This includes mean sea level pressure ($p$), 2 m air temperature ($T$), wind speed in 10 m ($v$), wind direction in 10 m ($d$) and precipitation amount ($RR$). Using $T$ and $p$, we further compute the potential temperature ($\theta$). Our focus is on Europe, more specifically stations within

the area of $-10\,°E$ to $20\,°E$, $40\,°N$ to $60\,°N$. Around 1700 stations are included of which, however, less than 400 stations on average observe all five parameters. For the training of the RF (Sect. 4), we focus on stations that measure at least three of the five parameters. The most frequent missing parameter in the hourly data is $RR$, as many stations only measure 3- or 6-hourly precipitation. However, many stations, especially over Germany, measure $RR$ only and, hence, are not usable for the training of the RFs (Sect, 4.2) but still helpful to inform our subjective labelling (Sect. 4.1). In addition, we exclude mountain stations,

that is, those with a station height above $800\,m$, as we suspect these to be dominated by orographic influences that may blur the feature characteristics we want to identify. This leaves around 750 stations per time step.

In order to take into account the diurnal and seasonal cycles as well as location-specific characteristics (e. g., exposed stations in coastal regions) in $\theta$ and also $v$, we decided to normalise these parameters by their climatology. For $\theta$, this means $\tilde{\theta} = \theta/\theta_{50}$, where $\theta_{50}$ is the median for the specific location, time of day and day of the year $\pm10$ days. This is done analogously for $v$

using the $98^{\text{th}}$ percentile: $\tilde{v} = v/v_{98}$, as we are mostly interested in high winds in this work. Both $\theta_{50}$ and $v_{98}$ are computed for the time period 2001 to 2019. Moreover, we are interested in temporal tendencies of $p$, $\tilde{\theta}$ and $d$, here represented simply by the difference between the current and the prior time step ($\Delta p$, $\Delta\tilde{\theta}$ and $\Delta d$, respectively). All parameters and their descriptions are listed in Table 1.

### 3.2 COSMO-REA6

As an example for a gridded data set, we use COSMO-REA6 data from the Hans-Ertel-Centre for Weather Research, which is a reanalysis based on the COSMO model from the German Weather Service (DWD) covering the European CORDEX domain with a grid spacing of $0.055\,\text{deg}$, i. e., roughly $6\,\text{km}$ (Bollmeyer et al., 2015). Several parameters, including wind, temperature,





**Table 1.** Overview of the variables considered for the objective identification using the probabilistic RF. The third column indicates whether the variable is used as a predictor for the final version of the RF. The associated percentiles and medians are computed with respect to the location, time of day and day of the year $\pm 10$ days.

| Variable | Description | RF | Derivation |
|---|---|---|---|
| $v$ | 10m wind speed | - | Station observation. |
| $\tilde{v}$ | Normalised wind speed | ✓ | $\tilde{v} = v/v_{98}$, where $v_{98}$ is the assoc. 98$^{\text{th}}$ percentile. Consider only $\tilde{v} \geq 0.8$ |
| $d$ | Wind direction | ✓ | Station observation |
| $\Delta d$ | Tendency of wind direction | ✓ | $\Delta d = d - d_{-1}$, where $d_{-1}$ is the observation of the previous hour |
| $p$ | Mean sea level pressure | ✓ | Station observation |
| $\Delta p$ | Tendency of mean sea level pressure | ✓ | $\Delta p = p - p_{-1}$, where $p_{-1}$ is the observation of the previous hour |
| $T$ | 2m air temperature | - | Station observation |
| $\theta$ | Potential temperature | - | Derived from $T$ and $p$ |
| $\tilde{\theta}$ | Normalised potential temperature | ✓ | $\tilde{\theta} = \theta/\theta_{50}$, where $\theta_{50}$ is the assoc. median |
| $\Delta\tilde{\theta}$ | Tendency of norm. pot. temperature | ✓ | $\Delta\tilde{\theta} = \tilde{\theta} - \tilde{\theta}_{-1}$, where $\tilde{\theta}_{-1}$ is the derived value of the previous hour |
| $RR$ | Precipitation | ✓ | Station observation |

humidity and pressure, are assimilated using a nudging technique. Observations from radiosondes, aircraft, wind profilers and surface stations are used for the nudging. However, precipitation is not assimilated, which can lead to larger deviations between

the reanalyis and observations by raingauges or radar (Bach et al., 2016; Hu and Franzke, 2020). The reanalysis is available from 1995 to 2019. This means that one of our case studies, namely storm *Sabine*, is not included (Table 2). The data set contains $p$, $T$, $RR$ and the zonal and meridional wind components, from which we can compute $v$ and $d$. Again, we further calculate $\tilde{\theta}$ and the temporal tendencies $\Delta p$, $\Delta\tilde{\theta}$ and $\Delta d$. Due to computational cost, we compute $\theta_{50}$ and $v_{98}$ for the 10-year time period 2005 to 2015 only but this should have a negligible effect on the final outcome. The data, originally on a rotated

grid, are regridded to a latitude-longitude grid with a grid spacing of $0.0625°$, i. e., roughly $7\,\text{km}$.

## 4 Method

The objective identification of mesoscale high-wind features within European wintertime cyclones includes three steps described in the following subsections. First, we identify the features subjectively in surface observations in several selected case studies, such that each station is assigned to a specific feature. These labels are then used to train RFs for feature prediction on

the basis of a cross-validation approach. In a final step, we obtain forecasts on a grid by interpolating the predicted probabilities using a Kriging approach. For the COSMO-REA6 data, the features are identified analogously. Instead of training separate RFs, we apply the RFs trained on the surface observations. As the COSMO-REA6 forecasts are already grid-based, the Kriging step is obsolete. We will refer to our newly developed method as RAMEFI (RAndom-forest based MEsoscale wind Feature Identification).





## 4.1 Subjective labelling using an interactive tool

Given the sometimes unclear distinction between the high wind features of interest in realistic cases, we decided to base our algorithmic development on how experienced meteorologists would identify the features on the basis of a wide range of parameters and their evolution in time and space. To accelerate and facilitate the subjective labelling of high-wind features, we developed an interactive tool using the open-source data visualisation package `bokeh` for Python (Bokeh Development Team, 2021), where one can switch between various parameters in a graphical display and select an area to set labels using a mouse-controlled lasso tool. A screenshot is provided in the Appendix (Fig. D1). The chosen parameters, taken from station observations or gridded data, include $\tilde{v}$, $p$, $\Delta p$, $\tilde{\theta}$, $\Delta \tilde{\theta}$, $RR$, $d$ and $\Delta d$. All used parameters are independent of the location of the station/grid point and horizontal gradients such that, theoretically, the approach can later be applied to a single station. Furthermore, the approach is independent of temporal evolutions beyond 1 h.

To distinguish the features, the locations of the fronts and the cyclone centre are important, such that analysis charts of the DWD and the UK Met Office were used for orientation. For a more detailed analysis, we further used the 3D front detection of the interactive visualisation software Met.3D (Rautenhaus et al., 2015). We only label stations with a high wind speed $\tilde{v} \geq 0.8$, that is, reaching 80% of their 98[th] percentile.

The guiding principles for the labelling were extracted from the scientific literature (cf. Sect. 2). As the WJ occurs within the warm sector, we expect warmer temperatures in this region, mostly westerly winds and almost no precipitation. Furthermore, $p$ decreases ahead of the cold front (Lackmann, 2011). With the arrival of the cold front, $p$ starts increasing again, temperatures cool and we find a veering shift in wind direction. As the name implies, CFC is mainly marked by a line of (heavy) precipitation. The CJ occurs on the colder side of the cyclone near the cyclone centre during maximum intensity, such that $p$ is generally lowest for this feature. It is often described to have a hook-shaped footprint in surface wind along the tip of an occluded or bent-back front. An SJ is labelled, where model-based trajectories analogous to Eisenstein et al. (2020) confirm a descending airstream. The area behind the cold front that is not associated with the CJ or SJ is labelled as the CS. An example of the reasoning behind the labelling is described for one time step of storm *Burglind* in Sect. 5.

The subjective labelling was done by the first author for 12 extratropical cyclone case studies listed in Table 2. The selection was based on their Storm Severity Index (SSI) over Germany (see Appendix C), caused damage and impacted area, while overall attempting to capture a healthy diversity of cyclones and features. The selected cases occurred during the extended winter half year between the end of September and end of March. They vary in terms of their cyclone tracks (Fig. 2) and occurring high-wind features, and include both NC and SKC. Two case studies developed an SJ, namely Egon (Eisenstein et al., 2020) and Friederike, which are included in the CJ feature here as explained in Sect. 4. We also include two storms, named *Herwart* and *Sabine*, where high wind speeds are enhanced due to a strong pressure gradient, such that it is more difficult to distinguish the features and the contribution of them to the storm's wind footprint. Further, *Sabine* stands out to be an extremely deep cyclone with a minimum core pressure of 944 hPa during its lifetime. More information on can be found in Sect. 7. In total 282 time steps have been analysed.




**Table 2.** Selected winter storm cases from 2015 to 2020 over Central Europe (names as given by FU Berlin), date, maximum observed gust speed (location), SSI over Germany and associated high-wind features. The cyclone tracks are displayed in Fig. 2

| Case | Date | max. gust [$\mathrm{km\,h^{-1}}$] | SSI | Features |
|------|------|------------------------------------|-----|----------|
| Niklas | 31 Mar 2015 | 192 (Zugspitze, D) | 20.8 | WJ, CFC, CJ |
| Susanne | 09 Feb 2016 | 158 (Patscherkofel, AT; Pilatus, CH) | 3.6 | WJ, CFC, CJ |
| Egon | 12-13 Jan 2017 | 150 (Fichtelberg, D) | 5.9 | WJ, SJ, CJ |
| Thomas | 23-24 Feb 2017 | 158 (Brocken, D) | 3.0 | WJ, CJ |
| Xavier | 05 Oct 2017 | 202 (Snezka, CZ) | 6.3 | WJ, CJ |
| Herwart | 29 Oct 2017 | 176 (Fichtelberg, D) | 15.2 | WJ, CFC, strong pressure gradient |
| Burglind | 03 Jan 2018 | 217 (Feldberg, D) | 15.2 | WJ, CFC, CJ |
| Friederike | 18 Jan 2018 | 204 (Brocken, D) | 18.3 | WJ, SJ, CJ |
| Fabienne | 23 Sep 2018 | 158 (Weinbiet, D) | 4.6 | WJ, CFC |
| Bennet | 04 Mar 2019 | 181 (Cairngorn, UK) | 5.1 | WJ, CFC |
| Eberhard | 10 Mar 2019 | 194 (Snezka, CZ) | 10.1 | WJ, CJ |
| Sabine | 09-10 Feb 2020 | 219 (Cape Corse, FR) | 20.0 | WJ, CFC, strong pressure gradient |

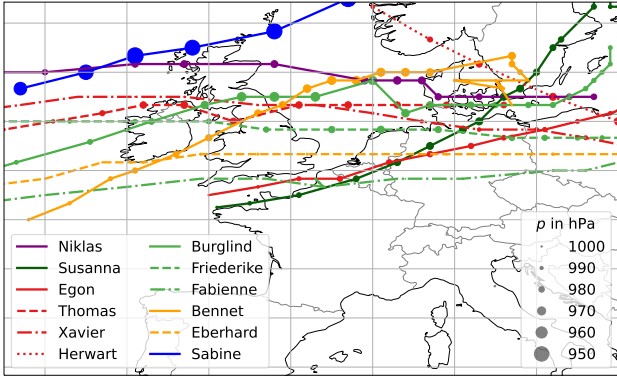

**Figure 2.** Cyclone tracks of case studies from Table 2. Colour indicates the year (purple = 2015, dark green = 2016, red = 2017, green = 2018, orange = 2019, blue = 2020), linestyle indicates the order in that year (solid - dashed - dashdot - dotted). The size of the markers correspond to the minimum mean sea level pressure $p$.

As mentioned above, we excluded mountain stations and stations where less than three of the given parameters were measured. This leaves around 750 stations per time step for the subjective labelling. Overall for the 12 case studies, we have 77.517 data points where $\tilde{v} \geq 0.8$, of which 19.200 (24.77 %) are not associated with a feature (NF), 19.501 (25.16 %) were labelled as a WJ, 3.800 (4.9 %) as CFC, 11.705 (15.1 %) as CJ, 1.502 (1.94 %) as SJ and 21.809 (28.13 %) as CS. Since the SJ is a small, short-lived and rare feature and the characteristics of SJs and CJs in surface parameters are very similar due to the proximity in both time and space, it is included in the more frequent CJ feature, increasing the values for CJ to 13.207 data points (17.04 %).





The features were further labelled in all case studies using the interactive tool for COSMO-REA6 data – except for *Sabine*
that occurred outside of the reanalyis time period. These labels are used to evaluate the predictions generated by the station-
based RFs for a grid-based data set (Sect. 4.2 and 6.2). For computational reasons, we downsampled the COSMO grid to every
third grid point in the zonal and meridional directions, resulting in a grid spacing of $0.1875 \, \mathrm{deg}$, i. e., around $21 \, \mathrm{km}$. Moreover,
we excluded ocean grid points, as the characteristics of the high-wind features might be different from land due to different
surface friction, surface heat fluxes, etc. Regions with high wind speed not directly associated with a winter storm, especially
over Italy and the Balkans, were not labelled.

## 4.2 Probabilistic Random Forest

An RF (Breiman, 2001) is a popular, non-parametric machine learning method for classification and regression problems that
does not rely on distributional assumptions but instead is based on the idea of decision trees (Breiman, 1984). A decision
tree subsequently splits the predictor space based on order criteria at its so-called nodes and thereby generates a partitioning.
The order criteria are chosen automatically such that the resulting subsets of observations associated with a partition are as
diverse as possible, where diversity is defined through a chosen splitting rule. The subspaces induced by the partitioning, the
so-called leaves, include a set of observations in the training set from which an analogue forecast is derived. In case of multi-
class probability forecasts, the class frequencies within the terminal leave are taken as forecast. An RF builds a randomised
ensemble of decision trees, where the generation of each tree is based on a different subsample of the data and that of each
node on a different subset of predictor variables. To obtain a final probability forecast from the ensemble of decision trees, the
individual predictions are aggregated. For further details on RFs, we refer to Breiman (2001) and Hastie et al. (2009).

Machine learning methods such as the RF are often referred to as black boxes due to a lack of interpretability, although
there exist several techniques to understand what the models have learnt and how the predictions are related to the predictors
(McGovern et al., 2019). We will apply two of these predictor importance techniques, one to find the most relevant predictors
and one that illustrates the effect of the predictor values on the forecast probabilities. The first is the permutation importance
of a predictor (Breiman, 2001). Proceeding separately for each predictor, the values within the test set are shuffled randomly
such that the link to the target variable is broken. Then, based on these permuted predictors, new predictions are generated
and compared to the predictions obtained with the original data. The worse the forecasts become (with respect to an evaluation
measure), the more important the predictor. Here, we measure forecast performance with the Brier score (BS; Brier, 1950). The
effect of a predictor on the predictions can be illustrated with a partial dependence plot (PDP; Greenwell, 2017). Given a fixed
predictor, a PDP shows the expected probability forecast as a function of the value of the predictor variable while averaging
out the effects of the other predictors. Hence, a PDP illustrates how the probability forecasts depend on the value of a specific
predictor variable, on average. For more details, we refer to McGovern et al. (2019).

In this study, we apply RFs to generate probabilistic forecasts of the wind features presented in Sect. 2. The parameters used
are listed in Table 1. As mentioned in Sect. 4.1, all parameters are independent spatially as well as temporally beyond $1 \, \mathrm{h}$. This
local approach gives us more flexibility but can cause detriments. Advantages and disadvantages of this approach are discussed
in Sect. 7.4. For the station-based observations, we use a cross-validation scheme on the different winter storm cases, that is,





for each winter storm the predictions are based on an RF that is trained on the data of the remaining 11 winter storms. Training RFs in a similar cross-validation scheme for the COSMO-REA6 data becomes computationally infeasible as the underlying
data sets become too large. As the underlying processes should coincide for both the station- and model-based data, we instead apply the station-based RFs in the same cross-validation scheme to generate probability forecasts using the COSMO-REA6 data. Details on the implementation including the choice of the hyperparameters can be found in the Appendix.

### 4.3    Kriging

As it is difficult to envision a coherent area of a certain wind feature from probabilities at single stations that are distributed
irregularly over the study area, we interpolate the station-based probability forecasts to a regularly spaced grid in order to visualise the results. In geostatistics, this is generally achieved by methods based on the best linear unbiased prediction (BLUP), commonly referred to as Kriging (Matheron, 1963). In principle, the Kriging predictions (here on the grid) are the weighted averages of the input data (here the station data), where the specification of the weights is driven by the covariance of the underlying random process. Under the assumption that the input data is a partial realisation of a Gaussian process (Rasmussen
and Williams, 2005), Kriging provides the optimal full predictive distribution. The key requirement for the implementation of Kriging in the context of Gaussian processes is the specification of the mean and the covariance function. Mathematical details regarding the Kriging approach taken here are provided in the Appendix.

In this study, we perform univariate Kriging to obtain probability maps for each wind feature, where we specify the mean and covariance function by a constant mean function and the stationary Matérn covariance function (Matérn, 1986; Guttorp
and Gneiting, 2006). For the estimation, we resort to the method of maximum likelihood estimation for Gaussian processes. However, since the input data are, in our case, multi-class probabilities and thus deviate from the Gaussianity assumption, we perform a data transformation for approximate Gaussianity. For the production of probability maps, we independently perform Kriging on each of the class probabilities, and execute probability sum normalisation on the predicted probabilities for each grid point such that, across multiple classes, the predicted probabilities sum to 1. Note that the Kriging predictions are only
obtained for areas over land, where our winter storms occurred and where a sufficient amount of data was available for a reliable interpolation. More details on the practical implementation of Kriging are also provided in the Appendix.

## 5    Illustrative case study: Storm *Burglind*

In this section a full case study for Storm *Burglind* is presented to illustrate the functionality of the new feature detection method RAMEFI. *Burglind* is relatively close to a "textbook" cyclone and shows a feature evolution largely in concordance
with the literature. Further examples and a general discussion on strengths and weaknesses of RAMEFI can be found in Sect. 7.



## 5.1 Synoptic evolution

Storm *Burglind* (int. *Eleanor*) developed as a secondary cyclone on 02 January 2018 over the North Atlantic and reached the British Isles at the end of that day. The core pressure dropped by more than $27\,\mathrm{hPa}$ in $24\,\mathrm{h}$ and, thus, exceeds the criteria for an explosive cyclogenesis after Sanders and Gyakum (1980). The minimum pressure occurred just east of the English North Sea coast around 03 UTC on 03 January. The cyclone then tracked mostly eastward across the North Sea and Baltic Sea before heading northeastward in later stages (Fig. 2). The cold front crossed France and Germany in the first half of 03 January and caused high winds due to CFC. Ahead of the front, high winds were associated with the WJ. Later, when the occlusion front wrapped around the cyclone centre, the CJ dominated as the cause of high winds in addition to CS winds further away from the cyclone core.

## 5.2 Application of feature identification

Figure 3 shows the most important parameters, namely $p$, $RR$, $\Delta\tilde{\theta}$, to distinguish the high-wind features for one selected time step, i. e., 03 January 2018, 06 UTC, to illustrate the process of the subjective labelling. In addition to the station data shown in Fig. 3, decisions for the labels were also aided by satellite imagery and synoptic charts (not shown). The set labels are shown in Fig. 3d. The cyclone centre was located to the east of the UK over the North Sea (red x in Fig. 3). The cold front stretched from northwestern Germany to France (see Fig. D4 in the Appendix). Highest winds were observed ahead of and along the cold front and over western France (contours in Fig. 3). As described in Sect. 4.1, $p$ decreases ahead of the cold front and starts to increase again afterwards. This can be seen in Fig. 3a with a strong decrease of mostly more than $3\,\mathrm{hPa}$ over southern and eastern Germany. The increase after the cold front is a lot weaker with some stations even still showing a weak decrease, which can for example be caused by small-scale processes like convection. Nevertheless, coinciding with the location of the front, several stations observe a $p$ increase of around $2\,\mathrm{hPa}$. This region also coincides with heavy rain with values of around $5\,\mathrm{mm\,h^{-1}}$ to more than $8\,\mathrm{mm\,h^{-1}}$ (Fig. 3b). Slightly lower amounts were observed along the occluded front and northern part of the warm front. Furthermore, we note a change in $\tilde{\theta}$ (Fig. 3c) with an abrupt decrease in the frontal region and a shift of $d$ from southwesterly to westerly winds (not shown). While $\Delta\tilde{\theta}$ indicates large tendencies over northern France and western Germany, it shows noisier behaviour further away from the highest winds. Following this, we set labels for a WJ in the region of negative $\Delta p$, positive $\Delta\tilde{\theta}$ and ahead of the high values of $RR$. CFC is labelled in the frontal region, where the heavy precipitation occurred. As signs for a CJ are missing at this point in time, the region behind the cold front is entirely labelled as CS. All set labels are displayed in Fig. 3d.

The forecasts of feature probabilities by the RF, which is trained on the 11 other cases (Table 2), for 06 UTC and other time steps in a $3\,\mathrm{h}$ interval are shown in Fig. 4 after Kriging was applied to generate a gridded field of probabilities. An animation for the entire lifetime of *Burglind* is provided in the supplementary material (Eisenstein et al., 2022b).

Comparing Fig. 3d and Fig. 4b shows the features mostly in consistent areas with high confidence. CFC is identified in a smaller region, which is partly due to missing precipitation observations, since this is the most important variable to predict CFC as will be discussed in Sect. 6.3.



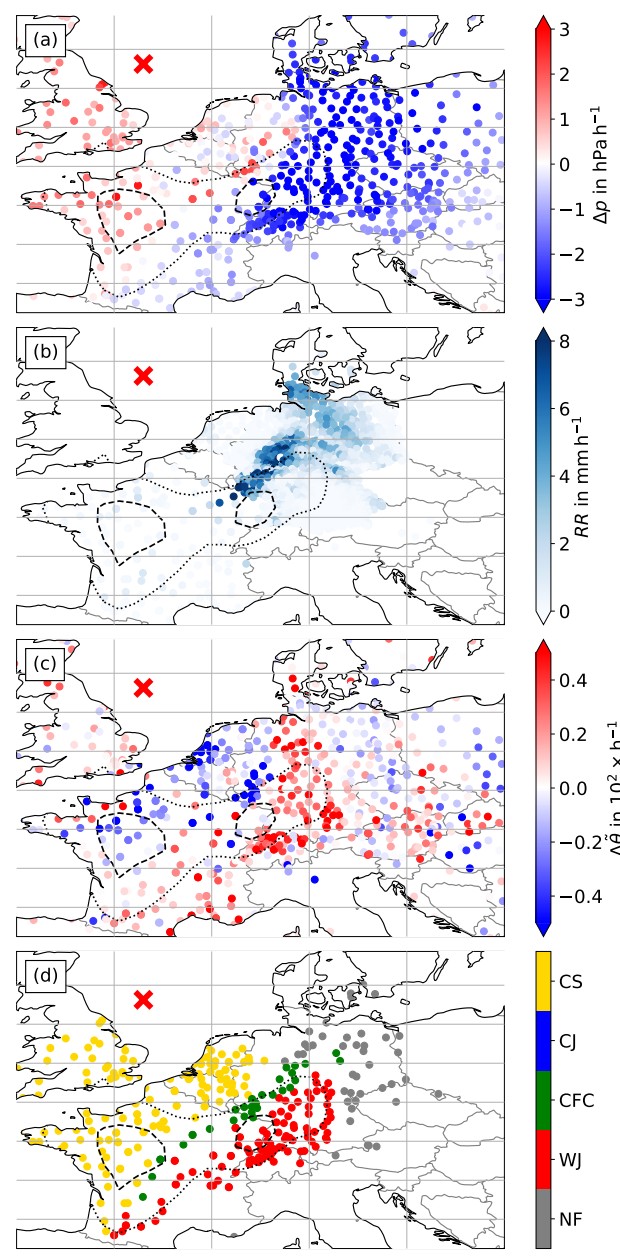

**Figure 3.** Storm *Burglind* on 03 January 2018, 6 UTC. Scattered dots show station observations for (a) $\Delta p$, (b) $RR$, (c) $\Delta\tilde{\theta}$ and (d) set labels for stations where $\tilde{v} \geq 0.8$ and at least three of the five initial parameters are measured (cf. Sect. 4.1). The contours show the interpolated $\tilde{v}$ (dotted $\tilde{v} = 0.8$, dashed $\tilde{v} = 1$, solid $\tilde{v} = 1.2$) to display the regions where the highest winds occurred. The red x indicates the cyclone centre

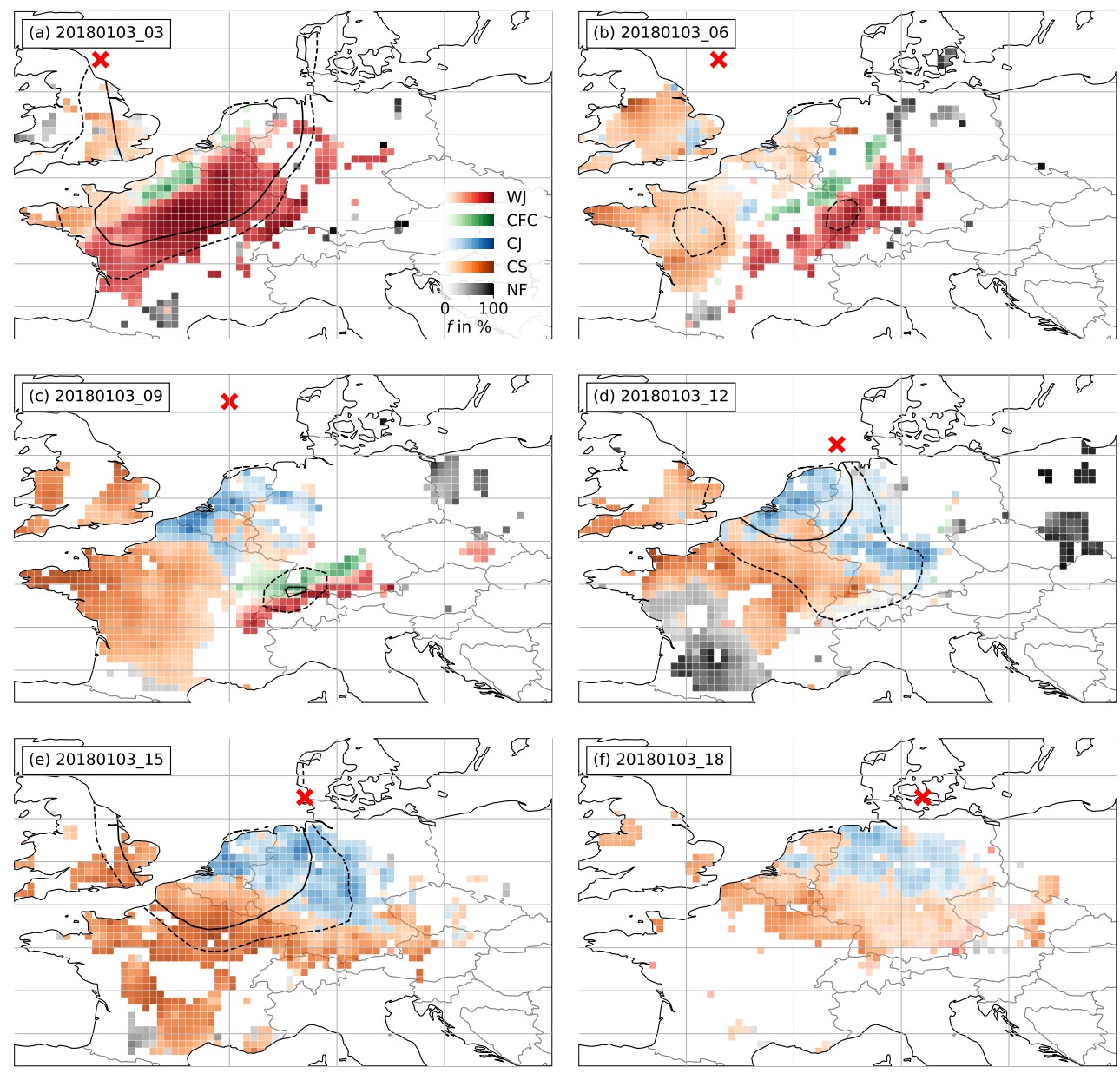

**Figure 4.** RF-derived probabilities $f$ for each high-wind feature after Kriging for Storm *Burglind* on 03 January 2018: (a) 03 UTC, (b) 06 UTC, (c) 09 UTC, (d) 12 UTC, (e) 15 UTC and (f) 18 UTC. The dashed contour shows $\tilde{v} = 1$, the solid contour $\tilde{v} = 1.2$). The red x indicates the cyclone centre.

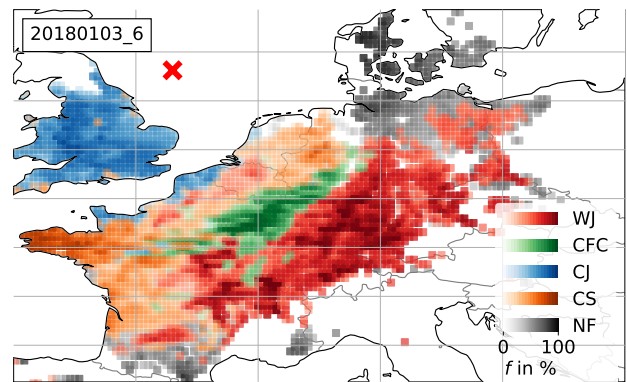

**Figure 5.** RF-derived probabilities $f$ for each feature for Burglind, 03 January 2018, 06 UTC as Fig. 4b but for COSMO-REA6.

As described in Sect. 2, the WJ is the first feature to occur during the lifecycle of a cyclone, as it is the case for *Burglind*. Figure 4a shows 03 January 2018, 03 UTC where high probabilities of a WJ are predicted for most of Central France. This region is followed by a smaller region of CFC along the front and CS behind it. As the cyclone evolves further, the cyclone centre and the identified features coherently move further east, while the area affected by the WJ diminishes (Fig. 4b). 3 h later the WJ dissolves north of the Alps, while still being followed by a line of identified CFC (Fig. 4c). At this time step, which

is also the time of minimum core pressure, the CJ is identified at the coasts of Belgium and the Netherlands, while the CS is detected further away from the cyclone centre. Highest $\tilde{v}$ are observed along the CFC and remainder of the WJ. The WJ and CFC vanish completely until 12 UTC, when the CJ and also the region of high $\tilde{v}$ extends to western Germany (Fig. 4d) and moves further east following the cyclone centre during the next 3 h (Fig. 4e). At 18 UTC, when the storm and $\tilde{v}$ weaken, the CJ starts to diminish and the probabilities of both CJ and CS decrease as well.

Even though the RF is only dependent on few meteorological parameters and their development over the last hour, looking at all time steps together, the features are largely coherent, both spatially and temporally, and behave as described in the literature (cf. Sect. 2). While the WJ and CFC appear in earlier stages of the lifecycle (Fig. 4a-c), they disappear in later stages, when CJ and CS dominate as the cause of high wind speeds. NF is mostly detected at the edges of the boundary or cyclone in all time steps, indicating that most of the high wind speeds in the vicinity of the cyclone are in fact associated with the introduced wind

features and identified accurately.

### 5.3   Comparison to gridded data

One interesting application of our new algorithm is the comparison of gridded forecasts with station observations. To do this successfully, we need to ascertain that such data sets can be fairly compared.

   To provide a visual impression of the differences between station- and reanalysis-based results, Fig. 5 shows the example of

*Burglind* again, such that it can be directly compared to Fig. 4b. Results are mostly consistent with even higher probabilities most of the time in the reanalysis. Particularly CFC covers a larger region with higher probabilities. This is due to the spatially




higher resolved $RR$ field in COSMO-REA6, the most important parameter for the detection of the CFC feature. Over the UK, high probabilities of a CJ are predicted by the RF. This contrasts with the identified CS in station data and the subjective labelling, where a CS was labelled, since no hook-shaped structure of high winds was discernible yet. This is consistent with the more common overforcast of CJ in the COSMO reanalysis than in observations as will be discussed in Sect. 6.2.

## 6    Statistical evaluation

In this section, we first describe how we generally evaluate the probabilistic forecasts generated by the RF and then apply this concept to the RF-based predictions for the station data and the COSMO reanalysis. At the end of the section, we investigate the relationship of the predictors and the predictions.

### 6.1    Assessing multi-class probability forecasts

Multi-class probability forecasts of three or more classes are typically evaluated by downscaling to two-class problems, of which the one-against-all and all-pairs approaches are two well-known examples (Zadrozny and Elkan, 2002). While the one-against-all approach compares the occurrence of one wind feature against all others grouped together, the all-pairs approach considers the conditional probabilities for each pair of classes, for example, the conditional probabilities of the WJ and the CJ when one of the two features materialises. The one-against-all approach is used to evaluate how well one specific wind feature is forecast, the all-pairs approach to evaluate the ability to discriminate between two wind features.

The binary predictions are evaluated based on the paradigm of Gneiting et al. (2007) that a forecast should aim to maximise sharpness subject to calibration. Calibration refers to the consistency of the forecast and the observation, while sharpness is a property of the forecast alone and refers to the associated uncertainty. In a nutshell, a binary probability forecast $f$ is called calibrated if the observed relative frequency (given $f$) matches $f$, for example, if a $20\,\%$ forecast is issued 100 times, the event should occur about 20 times. Further, a probability forecast is said to be sharper, the more certain it is, that is, the closer to 0 or 1. Both calibration and sharpness can be assessed qualitatively via reliability diagrams (Sanders, 1963; Wilks, 2011), which display the calibration curve that is close to the diagonal if the forecasts are calibrated. In addition to the calibration curve, the frequency of the probability forecasts is illustrated by a histogram. The more U-shaped the histogram is, the closer the forecasts are to 0 and 1 and thus the sharper. The reliability diagrams shown in this paper are based on the novel CORP-approach of Dimitriadis et al. (2021), which yields optimal calibration curves and eliminates the need of implementation decisions such as the number of bins that the calibration curve is based on. Quantitatively, calibration and sharpness can be assessed using the BS. Here, we compare our multi-class probability forecasts with a climatological benchmark using the multivariate Brier skill score. Details on the evaluation of multi-class and binary probability forecasts are provided in the Appendix.

### 6.2    Evaluation of the RF forecasts

The evaluation of the station-based RF forecasts is split into three parts. First, we quantitatively compare the RF forecasts with a climatological benchmark forecast, then we assess how well the RFs predict the individual wind features in the one-against-



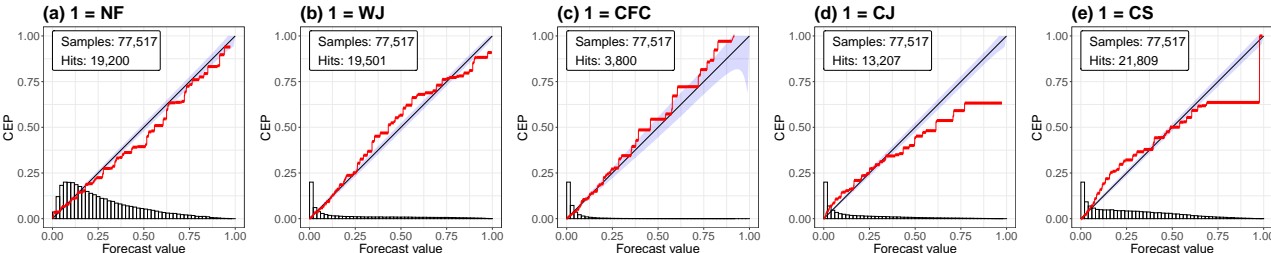

**Figure 6.** CORP reliability diagrams of the probability forecasts for the individual wind features in the one-against-all approach with uncertainty quantification via consistency bands (under the assumption of calibration) at the 90%-level.

all approach and finally we check how well the forecasts distinguish two features with the all-pairs approach. For each fold of the cross-validation scheme, the class frequencies within the training set are used as a climatological forecast. As expected, we

find that the RF forecasts outperform the climatology in terms of the multivariate BS for the prediction of each winter storm. The overall improvement is $24.7\,\%$, while for the different storms it ranges from $11.8\,\%$ to $34.7\,\%$ with $11.8\,\%$ being the skill for *Xavier*, which is discussed in some detail in Sect. 7.4.

Figure 6 shows the reliability diagrams of the binary probability forecasts in the one-against-all approach for the occurrence of NF and the four specific wind features (WJ, CFC, CJ and CS). We observe that the probability forecasts are in general

well-calibrated for all five cases, as the calibration curves closely follow the diagonal. The forecasts are generally reliable, especially for small probabilities, which are most frequent in this multi-class setting, as the peaks of the histograms illustrate. Therefore, the RFs identify the non-occurrence of a specific wind feature with high confidence (Fig. 6a). For larger forecast probabilities, the predictions of NF, the WJ and the CFC are well-calibrated, as the calibration curves stay reasonably close to the diagonal (Fig. 6a–c), while for the CJ and CS (Fig. 6d,e) larger deviations are evident. In both cases, the RF overforecasts

the events, that is, the predicted probability is generally too large.

The reliability diagrams of the all-pairs approach are displayed in Fig. 7, which show that the RFs yield well-calibrated forecasts for the distinction of all feature pairs but one. When the RF predicts that the CJ is more likely to occur than the CS (in case one of those two materialises), the RFs overforecast the CJ, meaning that the CS occurs more often than identified (Fig. 7j). This is consistent with the results from the one-against-all approach, where we found that the CJ and CS forecasts were not

well-calibrated for high probabilities, indicating that the RF fails to distinguish them for large conditional probabilities of the CJ. Further, the histogram of this pairwise comparison shows that the RF cannot discriminate between the two features with high confidence. This issue can be seen best for the storms *Herwart* and *Sabine*, which both did not develop a CJ, however a CJ was identified by the RFs (cf. Sect. 7.2). The main meteorological reason for this problem is the general similarity of the two features such that the distinction is mainly based on $p$, as will be discussed in Sect. 6.3. Other than that, the calibration

curves of the other pairs follow closely the diagonal. Moreover, we note that the WJ is distinguished well from the CJ and CS, as the U-shaped histograms of the forecast distributions show (Fig. 7f,g).



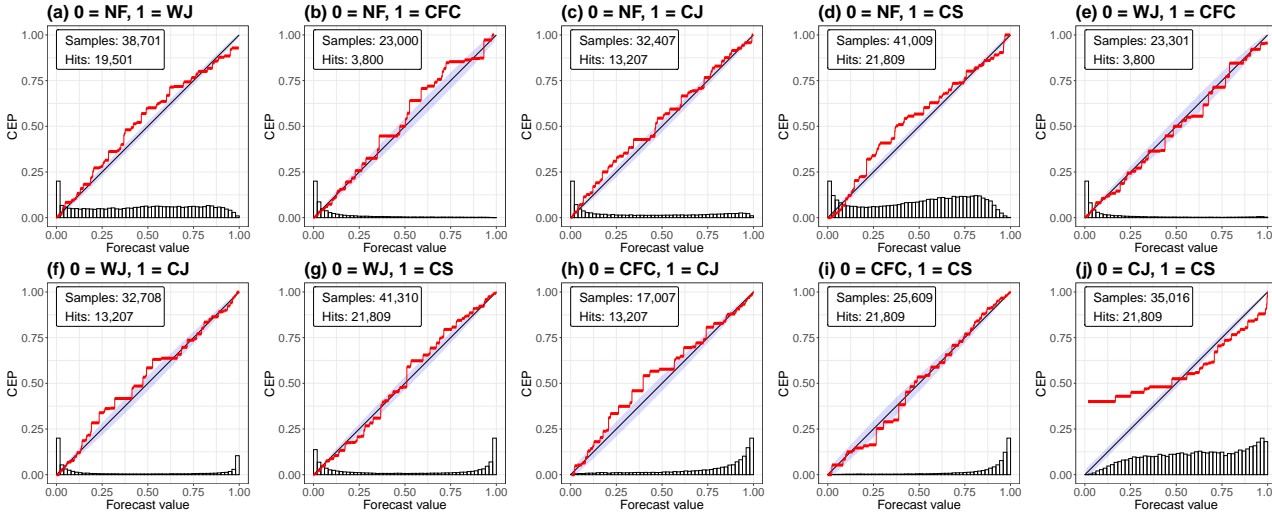

**Figure 7.** CORP reliability diagrams of the conditional probability forecasts comparing two wind features in the all-pairs approach with uncertainty quantification via consistency bands (under the assumption of calibration) at the 90%-level.

For the predictions derived from the COSMO-REA6 data, the RF forecasts are also able to distinguish the features well, although the used RFs were trained on station-based data. The forecasts exhibit similar characteristics and perform, as expected, only slightly worse than for the station data. As before, the skill of the multivariate BS is calculated with respect to

a climatological forecast, which for all storms is $19.6\,\%$. For eight of the selected storms, we observe improvements ranging from $11.0\,\%$ to $37.5\,\%$, however for *Herwart* and *Susanna* the skill scores are $-0.8\,\%$ and $-11.6\,\%$, respectively, indicating a decrease in forecast performance. For *Susanna*, this is due to a larger high-wind region ahead of but not directly connected to the cyclone for multiple time steps. While the predictions for *Herwart* look consistent in both data sets at first sight, less stations over Poland, where the CJ was overforcast (cf. Sect. 7.2), are available, such that the overforcast in the gridded data

carries more weight compared to the station data.

Further, we find at times high probabilities of mostly WJ in COSMO-REA6 in regions where winter storms are uncommon and where no features were labelled at all, as e. g., Italy and the Balkans (not shown). However, on the synoptic scale, the trough still affects some parameters in the region, that is, decreasing $\Delta p$ on the eastern side of the core and $d$ as the wind follows the isobars (not shown), which are the most important parameters to distinguish NF and WJ (Fig. 9). Therefore, high

winds caused by, e. g., mountainous effects, such as the foehn effect or land-sea breeze, might be falsely identified as a WJ. Thus, the RF should only be applied to regions affected by extratropical cyclones. As these regions have not been labelled (cf. Sect. 4.1), we excluded them from our evaluation.

The reliability diagrams of the one-against-all approach for the COSMO-REA6 data (see Fig. D2 in the Appendix) show that the calibration curves deviate more from the diagonal than for the station-based data (Fig. 6) but are still reasonably close

to calibrated. For the WJ and the CJ, we observe slight overforecasting (Fig. D2b,d), whereas we observe underforecasting for





the CFC (Fig. D2c). For the CS, we observe a similar calibration curve to the station-based data (Fig. D2e). The distinction

of the individual features, which we assess via the all-pairs approach in Fig. D3, results in mostly well-calibrated forecasts.
The largest deviations from calibration are observed again for the distinction of the CJ and the CS, as discussed above, and for
the distinction of the WJ and the CFC (Fig. D3e), where the WJ is identified more frequently than observed. This might be
caused by a spatially extended area of precipitation further into the warm sector at times due to missing data assimilation for
that parameter (cf. Sect. 3.2). Overall, the predictions based on the COSMO-REA6 data are satisfactory considering that the
RF models were trained on data from the station observations.

### 6.3 Predictor importance

To identify the predictors most relevant for the prediction of the wind features and the discrimination between two features,
we calculate the BS permutation importance for the one-against-all and all-pairs approach. The BS permutation importance in
the one-against-all approach is displayed in Fig. 8. In general, $\Delta p$ is the most important predictor variable, especially for the
WJ. Only for CFC it is not an important predictor, as it can occur slightly ahead of the cold frontal pressure trough, hence in a
region of positive $\Delta p$, as described in Sect. 2.4. On the other hand, the absolute $p$ values seem to be of less importance for WJ
and NF, which occur further away from the cyclone centre than CJ and CS, for which $p$ indicates the proximity to the cyclone
centre. For CFC, we find instead that $RR$ is the most relevant predictor variable as expected, while being less important for
WJ, CJ and CS. $d$ seems to be relevant for most features, as it is a characteristic for the location relative to the cyclone centre.
This also leads to a high importance for NF occurring more frequently north or west from the cyclone centre. However, $d$ is
not important for CFC, probably as convection leads to a more variable wind direction and due to the characteristic jump in $d$
at cold fronts. To the contrary, $\Delta d$ is of minor relevance for all features as well as $\Delta \tilde{\theta}$. A more important temperature-based
predictor seems to be $\tilde{\theta}$, although again being less relevant for CFC. Lastly, $\tilde{v}$ shows its highest importance for NF, as higher
wind speeds are less likely to be found at the boundary of a cyclone.

In the all-pairs approach (Fig. 9), we can attribute the importance of the predictor variables more accurately. The key to
distinguish the WJ from all other features is $\Delta p$, especially from the CJ and CS. This is consistent with the one-against-all
discussion above. The large outlier in $\Delta p$ in WJ vs. CJ is related to storm *Herwart* as further discussed in the following section.
Of secondary importance is $d$, particularly when compared to CJ, CS and NF. Temperature also plays some smaller role in the
distinction of the WJ. For CFC the by far most important predictor is $RR$ but when compared against the CJ $p$, $\Delta p$, $\tilde{\theta}$, $\Delta \tilde{\theta}$
and $d$ also contribute. The positive outlier in $RR$ is related to storm *Fabienne* (not shown). The distinction of the CJ to other
features is more complex. $p$ is relevant in all CJ-pairs, as already discussed. The distinction of CJ from NF additionally hinges
upon $\Delta p$, $\tilde{\theta}$ and $d$

The shortcomings of the RFs to distinguish CS and CJ is also reflected in Fig. 9 by partly negative values for $p$, $\Delta p$ and
$\tilde{\theta}$. A negative value indicates that the RF forecasts perform better, when we break the link to the target variable by randomly
permuting the predictor values. As discussed further in the following section, this is mostly due to storm *Sabine*, which reached
an unusually low minimum core pressure of less than $950\,\mathrm{hPa}$ over the Norwegian Sea (cf. Fig. 2). Because of this, $p$ values
in the CS over continental Europe were similar to values typical of a CJ.



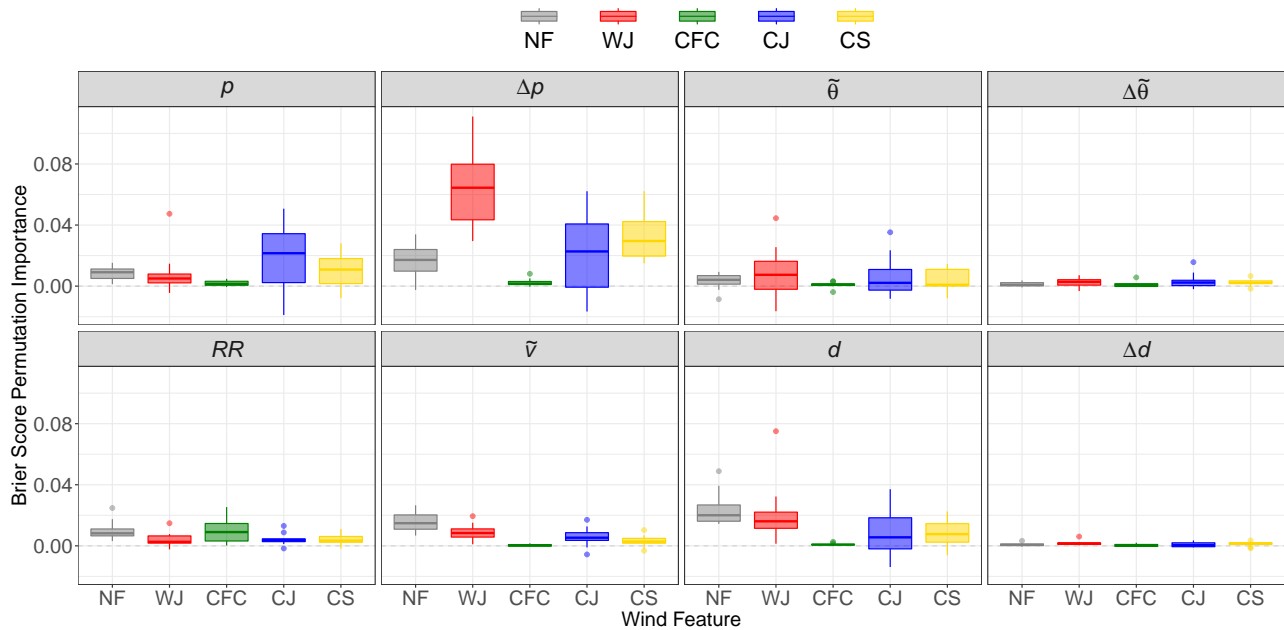

**Figure 8.** Boxplots of the Brier Score permutation importance of the probability forecasts for the individual wind features and predictor variables in the one-against-all approach. The boxplots are calculated over the folds of the cross-validation, that is, the individual winter storms.

We do not only want to identify the most relevant predictors, but also investigate their effect on the predictions, which is illustrated for the eight predictor variables by the PDPs in Fig. 10. Again, the largest impact is found for $\Delta p$. The probability of observing a WJ is largest for small values of $\Delta p$ and declines rapidly as the tendency increases and switches signs, while the probabilities of the CS and CJ increase. Probabilities for NF decrease slightly, while changes for CFC are small. For little $RR$ the probability of a CFC is close to zero, but consistently increases with increasing precipitation. In turn, probabilities for other features slightly decrease with increasing precipitation. In general, CJ and CS show high probabilities for low $p$ values consistent with their occurrence during the most intense stage of a cyclone (cf. Sect. 2). However, surprisingly CS shows higher probabilities than CJ between $970\,\mathrm{hPa}$ to $980\,\mathrm{hPa}$, although the CJ is usually closer to the cyclone centre. This is again associated with the unusual behaviour of storm *Sabine* with its deep pressure minimum but no subjectively identified CJ. As such intense cyclones are rare, we are confident that the RF performs well in most more ordinary cases. As discussed previously, $d$ is dependent on the location relative to the cyclone centre. As the introduced features are all located south to west of the cyclone, we focus on values from $90°$ to $360°$ only. Within the warm sector $d$ values mostly show south-westerly winds and do not change drastically. Probabilities for CFC increase with a positive wind shift, leading to more westerly and north-westerly winds for CFC but also following features, i.e., CJ and CS. $\Delta d$ shows almost no change in probabilities for all features consistent to its low BS permutation importance. $\tilde{\theta}$ shows an increasing trend for the WJ, while the probabilities decrease for the other features, most strongly for the CJ, as one would expect. For $\Delta\tilde{\theta}$ we see indications of the air mass change





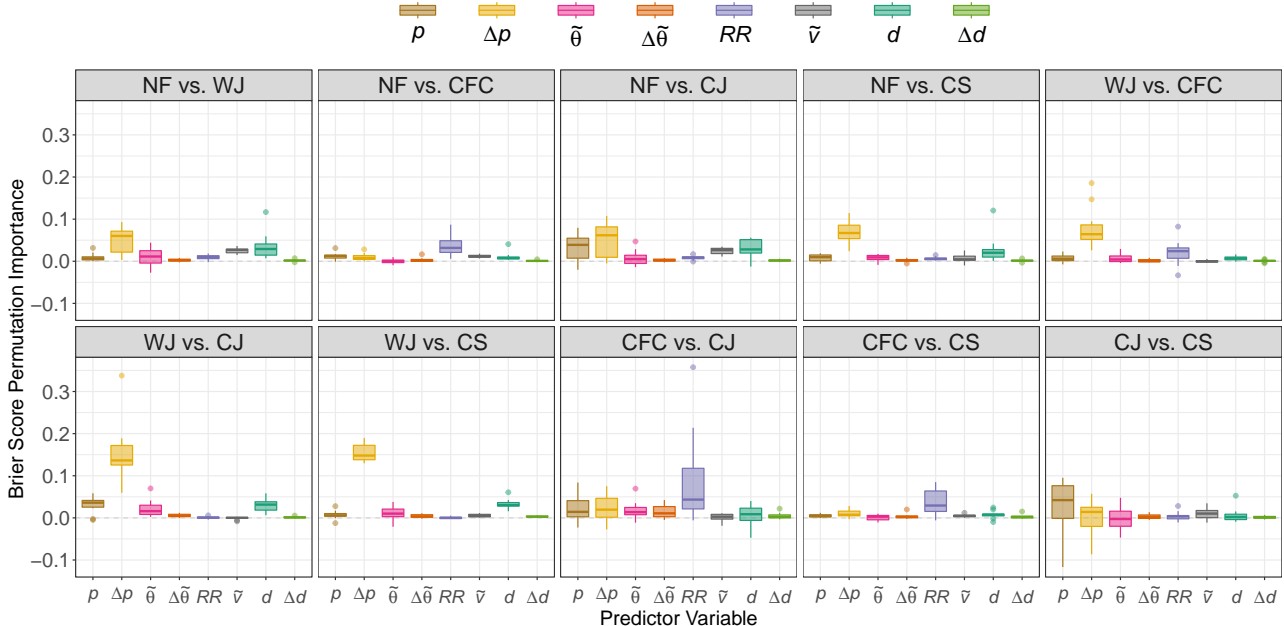

**Figure 9.** Boxplots of the Brier Score permutation importance of the probability forecasts comparing two wind features for the predictor variables in the all-pairs approach. The boxplots are calculated over the folds of the cross-validation, that is, the individual winter storms.

at the cold front and thus higher probabilities in CFC for negative values. The CJ shows a slightly positive trend, while all the others are flat.

Overall, investigating the importance of the predictor variables on the predictions, we find that the RFs largely learn physically consistent relations as described in Sect. 2 and Sect. 4.1.





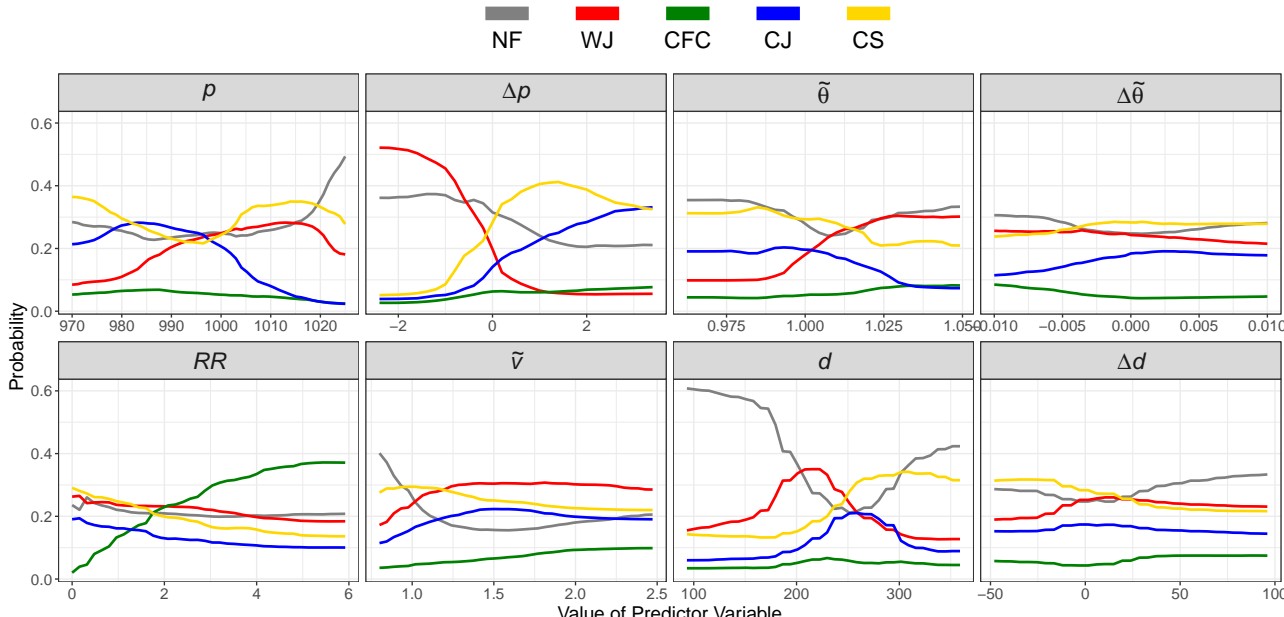

**Figure 10.** Partial dependence plots for the predictor variables and wind features.

## 7 Discussion

Section 5 showed a successful application of the introduced method to storm *Burglind* both in station- and grid-based data sets. However, extratropical cyclones rarely follow the textbook NC model exactly. For example, sometimes high winds are related to a strong synoptic-scale pressure gradient rather than associated with the mesoscale features that we have developed an objective identification algorithm for.

As described in Sect. 4, we selected 12 windstorm case studies to train the RF based on surface observations with the diversity of cyclones and features in mind, such that the RF is representative for a climatology over a longer time period. General information about these storms can be found in Table 2, the cyclone tracks and core pressure evolutions are shown in Fig. 2.

This section discusses how the trained RF deals with some well-known deviations from idealised cyclone models such as double fronts and convergence lines (Sect. 7.1), large background pressure gradients (Sect. 7.2), as well as the specific characteristics of SKC and SJ (Sect. 7.3). In addition, we further discuss advantages and disadvantages of not using spatial dependencies in the feature identification (Sect. 7.4). A complete set of results for all case studies can be accessed in the supplementary material (Eisenstein et al., 2022b).





### 7.1 Double fronts and convergence lines

Real-world frontal structures of extratropical cyclones can differ considerably from idealised conceptual models, e. g., in terms of complex vertical structure, strong tilts, or a secondary frontal zone parallel to the main front. Here, we are interested in synoptic systems with double cold fronts and convergence lines with high winds. The area between a primary and secondary cold front can have characteristics of a warm and/or cold sector and thus high-wind features are predicted with higher uncertainty by the RF. This can also be the case for the area between a cold front and a convergence line.

The example from our selection of 12 cyclones that illustrates this best is *Bennet*. On 04 March 2019 at 12 UTC the cyclone centre was located over the North Sea to the west of Denmark (Fig. 11a). The primary front is located at the north-eastern border of Germany and Bavaria and has already weakened (see Fig. D5 in the Appendix). A secondary strong temperature gradient can be found over northwestern Germany, Luxembourg and France. However, it is uncertain, if this should be classified as a front or convergence line. While synoptic charts from DWD show a convergence line, Met Office charts show an upper-level

cold front at 06 UTC and an occluded front 6 h later. As this feature shows characteristics typical of CFC (cf. Sect. 2.4), it was ultimately labelled as such by the first author.

Figure 11a shows the mesoscale wind features identified by the RF. At this time *Bennet* causes rather weak wind speeds, indicated by the small area of $\tilde{v} > 1$ (see dashed line in Fig. 11a). For the area between the fronts, the RF predicts both CS and WJ with medium confidence. This is to be expected, as some parameters, especially $\Delta p$, $\tilde{\theta}$ and $\Delta\tilde{\theta}$, show behaviour of both

features with a tendency towards CS. RF predictions along the secondary front/convergence line show only low probabilities for CJ and CFC. However, at earlier and later timesteps, that is, ahead of the primary cold front and behind the secondary one, the prediction of the WJ, CJ and CS are accurate, as can be seen in the full animation provided in the supplementary material (Eisenstein et al., 2022b). So looking at the entire lifetime of the storm, satisfactory identification can be obtained from RAMEFI.

### 510 7.2 Strong background pressure gradients

Very intense cyclones are often accompanied by a strong large-scale pressure gradient (e. g., Fink et al., 2009), which in turn causes high wind speeds unconnected to one of the four mesoscale wind features under study (or enhance the wind speeds associated with one of the features). Good examples from our list of case studies to illustrate this are the storms *Herwart* and *Sabine*. Figure 11b shows *Herwart* in a late development stage on 29 October 2017 at 09 UTC. Around this time, the cyclone

centre travelled over Poland (outside of the area shown in the figure). An occurrence of a CJ seems unlikely in that region, since a typical hook-shape structure cannot be seen in wind observations (not shown). The occluded front was rather weak and did not fully wrap around the cyclone centre (see Fig. D6 in the Appendix). So ultimately the highest wind speeds are enhanced by a strong background pressure gradient over Germany (black lines in Fig. 11b), making the subjective labelling quite challenging. The RF shows high probabilities of a CJ for several hours in this region, although it was originally labelled

to be CS. The main reason for this is that the proximity to the cyclone centre, reflected in low $p$, is the most important predictor

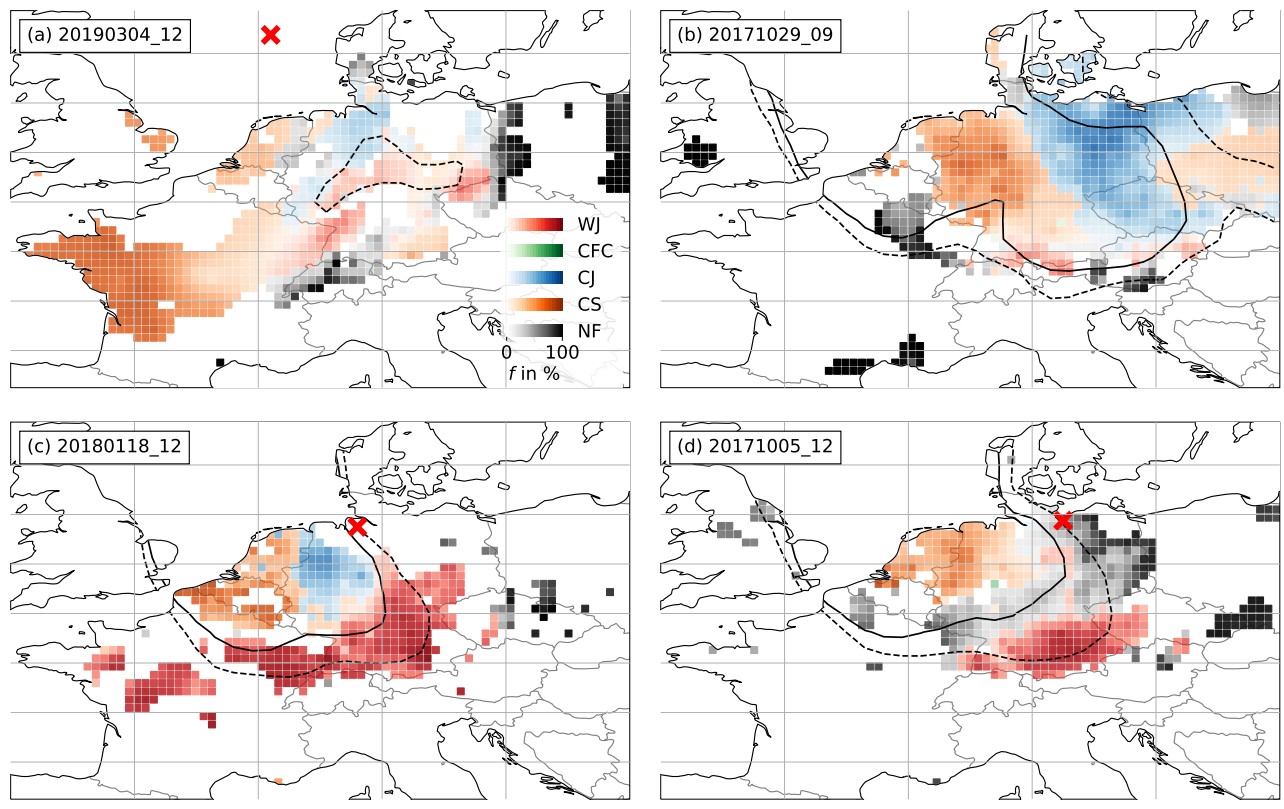

**Figure 11.** As Fig. 4 but for (a) storm *Bennet*, 04 March 2019, 12 UTC; (b) storm *Herwart*, 29 October 2017, 09 UTC; (c) storm *Friederike*, 18 January 2018, 12 UTC; and (d) storm *Xavier*, 05 October 2017, 12 UTC. Note that at the time shown, storm *Herwart* had already exited the plot area to the east.

to distinguish the CJ and CS (Fig. 9). Nevertheless and even in this unusual case, the prediction by the RF is still reasonable, since both a subjective and objective identification of the two features here is ambiguous in surface observations alone.

In the case of *Sabine* the cyclone centre did not cross continental Europe but moved through the Norwegian Sea (Fig. 2). The minimum pressure reached less than $950\,\mathrm{hPa}$. Stations over Central Europe still observed $p$ below $970\,\mathrm{hPa}$, which is lower
than the cyclone centres of most of our other case studies, making *Sabine* a quite unusual case. As discussed already in Sect. 6, this causes difficulties to distinguish the CJ and CS of *Sabine*, somewhat similar to *Herwart*. Although a CJ is identified in an area of low $p$ values, this region is not in the vicinity of the cyclone centre, as it was the case for *Herwart*. An animation of the feature identification for all time steps of *Sabine* is provided in the supplementary material (Eisenstein et al., 2022b). In this case the CJ-CS distinction issue could have been avoided to some extent by including spatial dependencies in the identification
algorithm at the expense of losing the capability for flexible application as discussed in Sect. 4. We will return to this issue in Sect. 7.4.





### 7.3 Shapiro-Keyser cyclones and sting jets

As already explained in Sect. 2, cold fronts of SKC are usually weaker than those of NCs, such that CFC wind features hardly occur. A good example for this is storm *Friederike*. Figure 11c shows 12 UTC on 18 January 2018, when the cyclone centre just reached the North Sea coast of Germany. High probabilities of a WJ occur over Central Germany, while a CJ is identified by the RF over northeastern Germany. CFC probabilities are very low along the entire cold front. In the western part of the high wind area (see black lines in Fig. 11c) a CS feature is detected. Lagrangian trajectories confirmed an SJ in the region where the CJ is identified during that time. As the CJ and SJ are considered together in this work (cf. Sect. 4.1), the area was labelled as CJ and, hence, identified accurately. This shows that CJ and SJ indeed show similar characteristics in surface parameters and can be considered as one feature in this context.

### 7.4 Spatial independence

The decision not to use spatial (nor temporal beyond 1 h) dependencies in the identification algorithm makes our method highly flexible in its application but the local approach can also cause issues where features deviate from their stereotypical characteristics. One example for the problem is the CJ of *Sabine* as discussed in Sect. 7.2. Another example is storm *Xavier*, where for several hours many points within the vicinity of the cyclone show the highest probability for NF rather than for any of the mesoscale wind features (Fig. 11d). The main reason for this appears to be that *Xavier* was characterised by unusually cool $\tilde{\theta}$ and high $p$ (not shown), generally two of the most important parameters to distinguish features (Fig. 8). In such a case, normalising $p$ by the core pressure could help detecting the region close to the cyclone centre, while the warm sector can be detected more reliably by comparing $\tilde{\theta}$ to the mean state over Europe during the period of the storm. While one predictor behaving in an unusual way could be compensated, as e. g., in the case of *Fabienne* (cf. Eisenstein et al., 2022b), two anomalous behaviours unsurprisingly result in considerably greater uncertainty.

A possible solution to the issues described here on the basis of *Sabine*, *Xavier* and *Fabienne* is to not only regard anomalies from diurnal and seasonal cycles but also to include some kind of spatial background, e. g., by subtracting a spatial mean. However, such a step would bring its own set of problems. Any spatial mean would require an arbitrary decision about the considered area, which may vary greatly from cyclone to cyclone. Moreover, spatial means computed from surface observations are not representative due to the irregular spacing of the stations. Essentially, as the features identified by the RF still occur in the expected areas as described in Sect. 2, we conclude that a flexible local approach offers more advantages than disadvantages overall.

## 8 Conclusions

High wind and gust speeds can be caused by distinct mesoscale features within extratropical cyclones, which occur during different stages of the cyclone lifecycle, in varying regions relative to the cyclone centre and have distinctive meteorological



characteristics (e. g., Hewson and Neu, 2015). These differences likely imply differences in hazardousness, forecast errors and, hence, risk to life and property.

To better understand, monitor and predict these mesoscale features, we developed RAMEFI, a first-ever objective identifi-
cation method that is able to reliably distinguish the four most important features, that is, the WJ, the CJ, CFC and CS. The rare and often short-lived SJ is included in the CJ category, as their surface characteristics are often rather similar and 3D trajectories are required for a clean distinction (Gray et al., 2021).

The first step was to build a browser-based, interactive tool to subjective label surface stations over Europe for 12 selected winter storm cases between 2015 and 2020. Based on the outcome, we trained a probabilistic RF based on the eight predictors
$\tilde{v}$, $p$, $\Delta p$, $\tilde{\theta}$, $\Delta\tilde{\theta}$, $RR$, $d$ and $\Delta d$. Being independent of spatial behaviour or gradients and only requiring $1\,\mathrm{h}$ tendencies, the approach is very flexible and can be applied to single stations or grid points and various data sets with differing grid spacing. To obtain areal information from irregular station data, Kriging was applied on the station-based probabilities generated by the RF.

The trained RFs are generally well-calibrated. Merely, the distinction between CJ and CS is more challenging, since the two
features show similar characteristics in most parameters except for lower $p$ in the CJ being located nearer to the cyclone centre. Overall, the RFs learn physically consistent relations reflected in the importance of individual predictors. For example, while $\Delta p$ appears to be most important for WJ, CJ and CS, $RR$ is substantial for the identification of CFC.

A detailed analysis of the RF feature probabilities for the selected cases shows a high consistency with the subjectively set labels with only few disagreements, mostly in cases of large deviations from standard cyclone models. While the identification
of WJs has the highest confidence, the identification of CFC is least certain due to relatively few surface stations reporting hourly precipitation and thus less training data. Even the distinction between the relatively similar CS and CJ works well in most cases and time steps. In some cases, however, high probabilities of CJs are predicted by the RF in areas where no CJ was identified subjectively due to a missing hook-shaped structure and occlusion front, or too large distance from the cyclone centre (e. g., *Herwart*, *Sabine*). Despite the spatial independence of the method, putting the predicted probabilities together on
a horizontal map and following the storm evolution in time shows a high degree of coherence for each feature, demonstrating the success of our method.

The station-based RFs are also applied to COSMO reanalysis data without any adaptations to the new data set. Nevertheless the obtained results are mostly consistent and only slightly less calibrated. This demonstrates that the method should be readily applicable to other analysis and forecast datasets.

Now that the RAMEFI method is fully developed and tested, it enables a number of exciting follow-on studies (see Part II). The next step we plan is to use the objective identification approach to compute a long-term climatology over Europe based on station observations and COSMO reanalysis data. Although, previous literature discussed different causes of winds within extratropical cyclones, their climatologies were based on more subjective categorisations for a limited sample size (e. g., Hewson and Neu, 2015; Earl et al., 2017). RAMEFI will for the first time allow a statistically substantiated analysis of
the characteristic of the mesoscale wind features in terms of size, lifetime, position relative to the cyclone core, occurrence relative to the lifecyle of the cyclone and wind characteristics. Furthermore, a systematic forecast error analysis will reveal to

what extent forecast errors differ between the identified features and whether there any significant, systematic deficits in their representation in models. Based on the outcome we plan to subsequently work on a feature-dependent postprocessing approach using the methods discussed in Schulz and Lerch (2022). This can ultimately improve the forecasts of strong winds and gust,

and thus support the provision of better warnings regarding high wind gusts and of ultimately timely forecasts of the associated impacts.

*Code availability.* The interactive visualisation and labelling tool, as well as the trained RF and Kriging code are available at https://gitlab. physik.uni-muenchen.de/Lea.Eisenstein/ramefi, where it will be updated in future studies, and is archived at https://doi.org/10.5281/zenodo. 6541303 (Eisenstein et al., 2022a).

*Data availability.* COSMO-REA6 data are available under https://reanalysis.meteo.uni-bonn.de (Hans-Ertel-Centre for Weather Research, 2019). Daily maximum gusts over Germany used for the computation of the SSI are available under https://cdc.dwd.de/portal/ (Deutscher Wetterdienst). Further observation data over Europe was provided by DWD for this work and cannot be made freely available. Please contact the DWD directly regarding this data

*Video supplement.* A video supplement showing the probability maps for all 12 selected case studies (Table 2) can be freely accessed at

https://doi.org/10.5281/zenodo.6541277 (Eisenstein et al., 2022b).

## Appendix A:  Implementation details

In this section, we provide technical details on the probabilistic RFs, the Kriging and the evaluation of the RF forecasts.

### A1   Random Forests

RF is implemented via the `ranger`-package (Wright and Ziegler, 2017) in R (R Core Team, 2021). Table 1 summarises the

predictors used, Table A1 the chosen hyperparameters. One question in the implementation is the handling of missing values, which an RF cannot process. The station-based samples frequently miss values of one or more predictor variables, especially precipitation is affected. We tried different strategies to handle missing values such as leaving out instances with missing values or replacing the missing values with a mean value and found similar results. Therefore, we decided to replace the missing values in order to use the largest sample size possible, which is desirable for the evaluation and the Kriging step. In each fold of the

employed cross-validation scheme, the missing values (both in the training and test set) are replaced by the mean value of the associated predictor variable in the training set.



**Table A1.** Overview of the hyperparameters of the probabilistic RF.

| Hyperparameter | Value |
|---|---|
| Number of trees | 1,000 |
| Number of predictors considered at each split | 2 |
| Minimal node size | 10 |
| Maximal depth | Unlimited |
| Splitting criterion | Gini |

## A2 Kriging

In our practical implementation of Kriging, we employ the R-packages: `fields` (Douglas Nychka et al., 2017), `mvtnorm` (Genz et al., 2021), `maps` (Original S code by Richard A. Becker and Allan R. Wilks. R version by Ray Brownrigg. En-

hancements by Thomas P Minka and Alex Deckmyn., 2018) and `maptools` (Bivand and Lewin-Koh, 2021). Additionally, we transform the probabilities by using the `bestNormalize` package (Peterson, 2021) to achieve approximate Gaussianity, which automatically chooses a suitable transformation from a set of commonly used transformations. The probabilities on the grid generated via the univariate Kriging need to be normalised such that they sum up to 1. However, at some grid cells distant from the cyclone track, the predicted probabilities are small for all of the wind feature and normalisation results in unrealistic

predictions. Thus, we only perform the normalisation at grid cells where the accumulated probability is larger or equal to 20 %. For the visualisation, we further drop the grid cells where the largest normalised probability is smaller than 20 % (which includes the grid cells for which no normalisation was performed).

## A3 Evaluation of the RF forecasts

The CORP reliability diagrams were generated using the `reliabilitydiag`-package (Dimitriadis et al., 2021), the PDPs

using the `pdp`-package (Greenwell, 2017). Each reliability diagram is based on the probability forecasts and observations of all storm cases. For the PDPs, one partial dependence curve has to be calculated for each RF generated in a fold of the cross-validation, that is, for each winter storm. The final curves are then obtained by a weighted average depending on the sample size of the folds.

## Appendix B: Mathematical details

In this section, we provide a more detailed description of the Kriging approach and the assessment of probability forecasts.





## B1 Kriging

Let $\{X(\mathbf{s}), \mathbf{s} \in \mathbb{R}^2\}$ be the spatial Gaussian process modelling the transformed probability of a certain wind feature, indexed by the spatial coordinates $\mathbf{s}$ that correspond to the latitude and longitude associated with the (transformed) probability. Further, we denote the mean function by $\mathbb{E}\{X(\mathbf{s})\} = \mu(\mathbf{s})$ and the covariance function by $\mathrm{Cov}\{X(\mathbf{s}), X(\mathbf{s}')\} = C(\mathbf{s}, \mathbf{s}')$. Then, for a
given set of station-based data $\boldsymbol{x} = \{X(\mathbf{s}_1), \ldots, X(\mathbf{s}_n)\}^T$, the spatial prediction at a grid cell $\mathbf{s}_0$ is given as $\hat{X}(\mathbf{s}_0) = \mu(\mathbf{s}_0) + \Sigma_{12}\Sigma_{22}^{-1}(\boldsymbol{x} - \boldsymbol{\mu})$, where $\boldsymbol{\mu} = \{\mu(\mathbf{s}_1), \ldots, \mu(\mathbf{s}_n)\}^T$, $\Sigma_{12} = \{C(\mathbf{s}_0, \mathbf{s}_1), \ldots, C(\mathbf{s}_0, \mathbf{s}_n)\}$ and $\Sigma_{22} = \{C(\mathbf{s}_i, \mathbf{s}_j)\}_{i,j=1}^n$. Additionally, one can obtain the prediction variance as $\mathrm{Var}\{\hat{X}(\mathbf{s}_0)\} = C(\mathbf{s}_0, \mathbf{s}_0) - \Sigma_{12}\Sigma_{22}^{-1}\Sigma_{12}^T$, and the full predictive distribution as $X(\mathbf{s}_0) \mid \boldsymbol{x} \sim \mathcal{N}\big(\hat{X}(\mathbf{s}_0), \sqrt{\mathrm{Var}\{\hat{X}(\mathbf{s}_0)\}}\big)$.

The choice and estimation of the mean function $\mu(\cdot)$ and the covariance function $C(\cdot, \cdot)$ are key elements of the Kriging
implementation. While one can choose any parametric or nonparametric functional representation for $\mu(\cdot)$, the valid choice for $C(\cdot, \cdot)$ is limited to the class of positive semidefinite functions. In practice, the covariance function is often assumed to be stationary, which implies that the covariance function depends on the spatial locations only through spatial lags, i. e., $C(\mathbf{s}_i, \mathbf{s}_j) = K(\mathbf{s}_i - \mathbf{s}_j)$ for some positive semidefinite function $K(\cdot)$. In our implementation, we have specified the mean function $\mu(\cdot) = c$, $c \in \mathbb{R}$ to be a constant valued function, and the covariance function $K(\cdot)$ to be the Matérn class of stationary covariance
function (Matérn, 1986).

## B2 Assessing multi-class probability forecasts

Consider a multi-class probability forecast $\mathbf{f} = (f_1, \ldots, f_k)$, where $f_1, \ldots, f_k \in [0,1]$ and $\sum_{i=1}^k f_i = 1$, for a nominal target variable $Y \in \{1, \ldots, k\}$ with $k \geq 3$ classes that are not ordered. In mathematical terms, $\mathbf{f}$ is called (auto-)calibrated if $\mathbb{P}(Y = i \mid \mathbf{f}) = f_i$ for all $i = 1, \ldots, k$, where $\mathbb{P}$ refers to the joint distribution of forecast and observation (Gneiting and Ranjan,
2013). The Brier score for a probability vector $\mathbf{f}$ and the realising class $i \in \{1, \ldots, k\}$ is given by $S(\mathbf{f}, i) = \sum_{j=1}^k (f_j - \mathbb{1}\{i = j\})^2$ (Brier, 1950). Now, consider a probability forecast $f \in [0,1]$ for a dichotomous target variable $Y \in \{0,1\}$. In mathematical terms, $f$ is called calibrated if $\mathbb{P}(Y = 1 \mid f) = f$ (Gneiting and Ranjan, 2013). The binary Brier score of a probability forecast $f$ and realisation $y \in \{0,1\}$ reduces to $S(f, y) = (f - y)^2$. Both the binary and multivariate Brier score can be used to evaluate the improvement of a forecast method over a reference using the Brier skill score. Given the mean Brier score of the
forecasts of interest $\bar{S}_f$ and that of the reference forecast $\bar{S}_{ref}$, the skill score is calculated via $SS_f = 1 - \bar{S}_f / \bar{S}_{ref}$. Positive skill indicates improvement over the reference with $100\,\%$ referring to a perfect forecast, $0\,\%$ to no improvement and a negative skill to a decrease in forecast performance with respect to the reference.

The one-against-all approach reduces the multi-class forecasting problem to a set of $k$ dichotomous problems. For each class $i \in \{1, \ldots, k\}$, the probability $f_i$ is a forecast for $\tilde{Y} = \mathbb{1}\{Y = i\}$, where $\mathbb{1}$ denotes the indicator function. Note that evaluating the forecasts $f_i$ for $\tilde{Y}$ for each class is not equivalent to checking the multi-class calibration criterion, as the joint distribution of $f_i$ and not that of $\mathbf{f}$ is considered in the one-against-all approach. The all-pairs approach reduces the multi-class forecasting problem to a set of $k(k-1)/2$ dichotomous problems. For each pair of classes $(i, j)$, where $i, j \in \{1, \ldots, k\}$ and $i > j$, we





consider only samples with $Y \in \{i, j\}$. Then, the conditional probability $\tilde{f}_{i,j}$ is a forecast for $\tilde{Y}$, where

$$\tilde{f}_{i,j} = \frac{f_i}{f_i + f_j} \quad \text{and} \quad \tilde{Y} = \begin{cases} 1 & \text{for} \quad Y = i, \\ 0 & \text{for} \quad Y = j. \end{cases}$$

## Appendix C: Storm Severity Index

The SSI was originally developed to estimate windstorm-related damage to buildings and infrastructure (Klawa and Ulbrich,
2003). With this aim, daily wind gust maxima ($v_{g,max}$) for DWD stations were first scaled with its $98^{th}$ percentile ($v_{g,98,s}$) to take
local conditions into account. Next, the exceedences above $v_{g,98,s}$ are cubed to account for the wind destructiveness and are
weighted with the population density as a proxy for the insured values. Later developments introduced formulations for grid-
based data, i. e., reanalysis data and climate models, definition of affected areas ("windstorm footprints") and feature tracking
(e. g., Pinto et al., 2007; Leckebusch et al., 2008). Following Pantillon et al. (2018), we use the SSI formulation for station
observations over Germany considering only the meteorological impact (no population weighting):

$$\text{SSI} = \sum_{\text{station s}} \left\{ \left( \frac{v_{g,max,s}}{v_{g,98,s}} - 1 \right)^3 \right\}_{v_{g,max} > v_{g,98}} \tag{C1}$$

Daily maximum wind gusts over Germany are available from the DWD surface network (Deutscher Wetterdienst). The SSI
is calculated for each single day and displayed in Table 2 for the 12 selected cases. If a storm affects the region for several
days, the maximum SSI of the daily values is selected. As the aggregated SSI value is dependent on the number of stations,
possible changes in the surface network need to be taken into account for when analysing longer time series. For the period of
our case studies (2015–2020), the number of stations remained stable, and thus a comparison of the SSI values is largely fair.
Still, the obtained SSI values only serve as a comparison for the selected case studies in this paper and should not be compared
numerically to SSI values from other works.





## Appendix D: Further figures

**Figure D1.** Screenshot of the interactive visualisation and labelling tool as part of RAMEFI (Eisenstein et al., 2022a) using the Python package `bokeh` (Bokeh Development Team, 2021). The user can switch between timesteps and loaded winter storms. Top row shows a map of stations and their parameters, here $\tilde{v}$, where the user can select an area by mouse using a lasso or clicking on single stations. Labels for the introduced features can be set for the selected stations. A table includes all data points and parameters. Histograms are shown for several parameters for the whole region (light blue) and currently selected stations (dark blue).





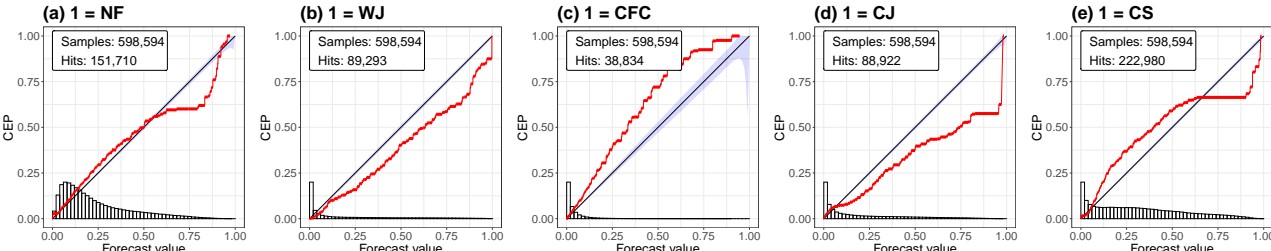

**Figure D2.** As Fig. 6 but based on COSMO-REA6 data.

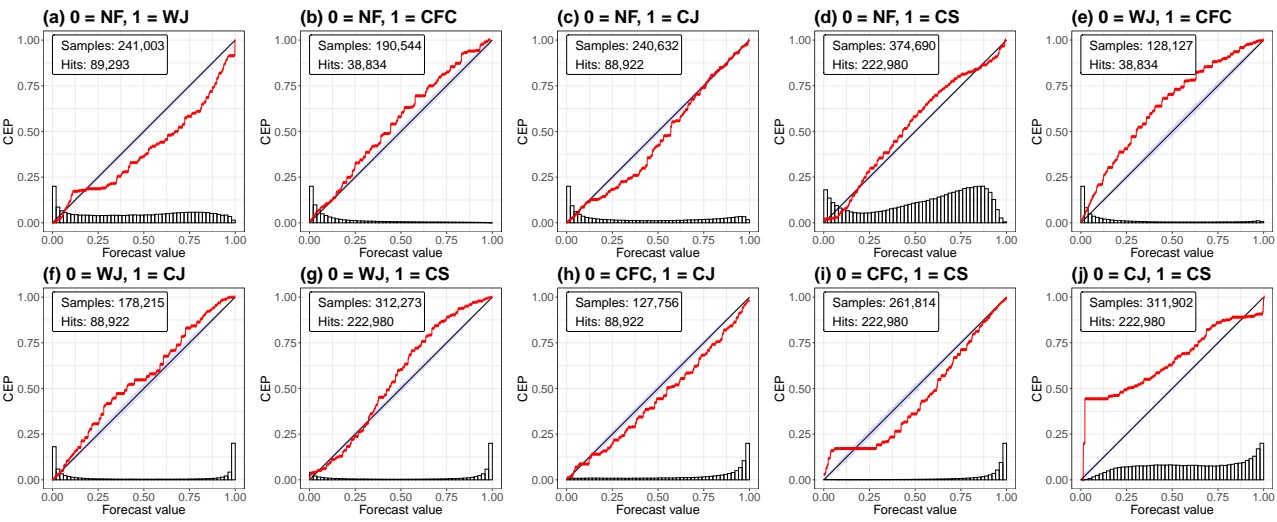

**Figure D3.** As Fig. 7 but based on COSMO-REA6 data.

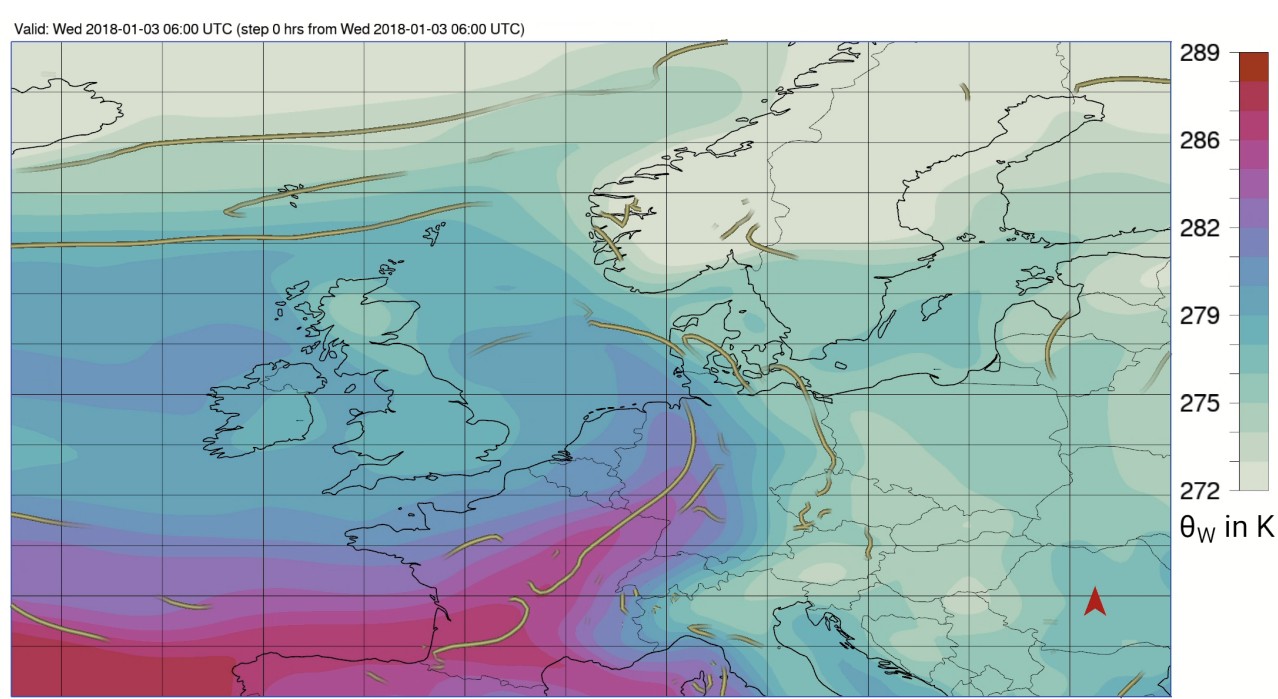

**Figure D4.** Detected fronts at $850\,\mathrm{hPa}$ (green tubes) and wet-bulb potential temperature $\theta_\mathrm{w}$ (shading) for storm *Burglind* on 03 January 2018, 06 UTC using the visualisation software Met.3D (Rautenhaus et al., 2015). Data source: ERA5.

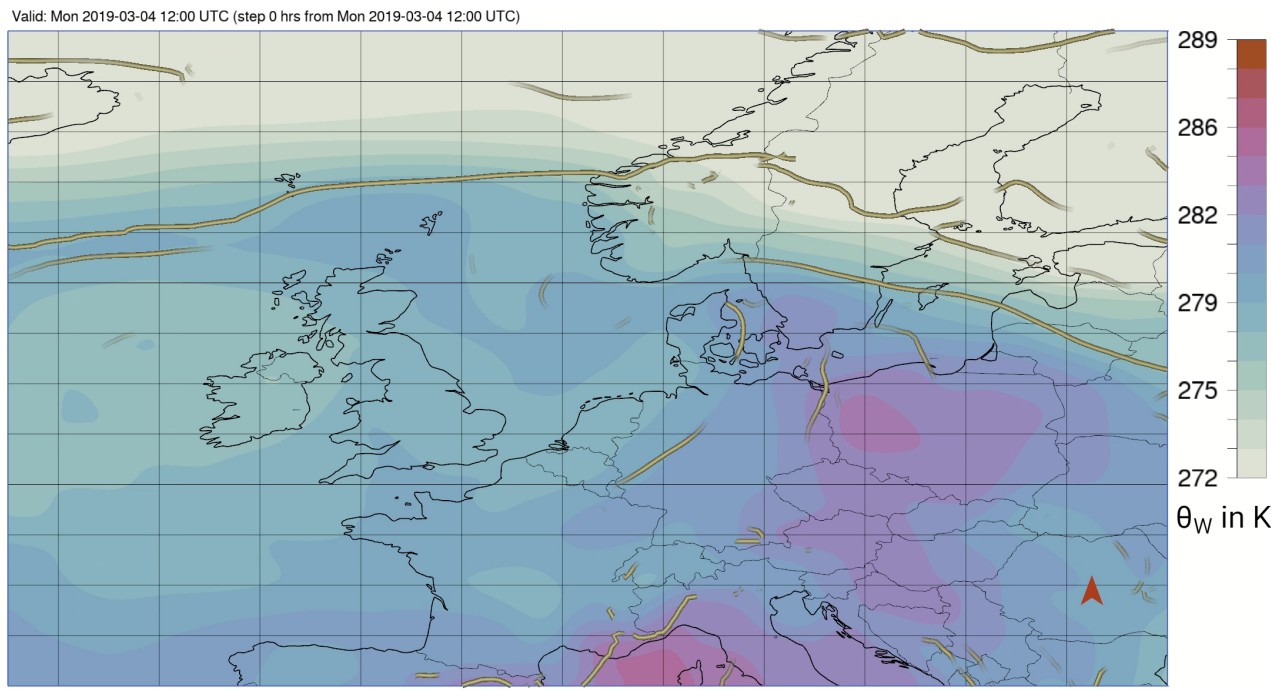

**Figure D5.** As Fig. D4 but for storm *Bennet* on 04 March 2019, 12 UTC.

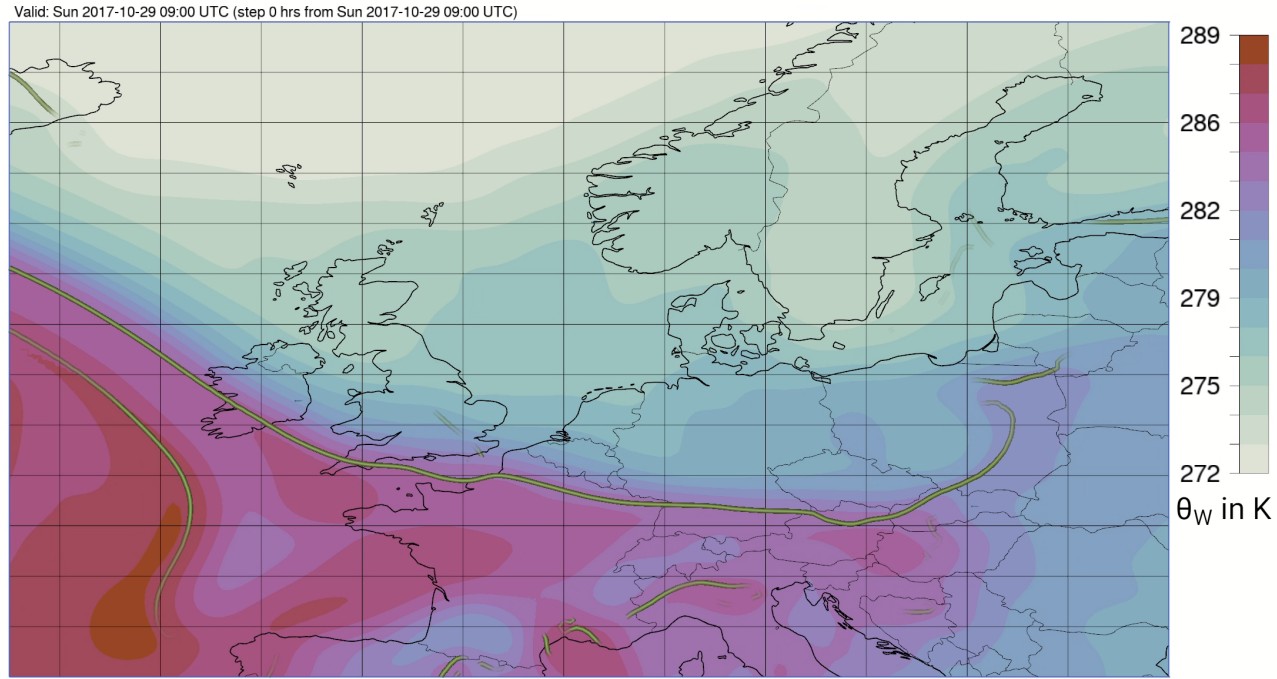

**Figure D6.** As Fig. D4 but for storm *Herwart* on 29 October 2017, 09 UTC,



*Author contributions.* LE developed the interactive tool, labelled the features, drafted the meteorological part of the study and wrote the original manuscript. BS applied the random forest, evaluated the results and drafted the mathematical part of the study. GAQ implemented the Kriging algorithm. PK designed the overall project, acquired the funding and coordinates the scientific work. PK and JGP jointly supervise the PhD of LE. All authors contributed with discussions and text revisions.

*Competing interests.* PK is member of the editorial board of *Weather and Climate Dynamics*. The peer-review process was guided by an
independent editor. The authors have no other competing interests to declare.

*Acknowledgements.* The research leading to these results has been accomplished within the project C5 "Dynamical feature-based ensemble postprocessing of wind gusts within European winter storms" of the Transregional Collaborative Research Center SFB/TRR 165 "Waves to Weather" funded by the German Science Foundation (DFG). JGP thanks the AXA Research Fund for support. We thank Sebastian Trepte and Olivier Mestre for providing the surface observations data set and Robert Redl for pre-processing the data. Further thanks go to Sebastian
Lerch and Johannes Resin for thorough discussions and helpful comments as well as Andreas Beckert for the front visualisation using Met.3D (https://met3d.wavestoweather.de). LE thanks Suzanne L. Gray for hosting her at the University of Reading for an extended research stay in fall 2021 and many discussions.



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
