# Peer review of "Identification of high-wind features within extratropical cyclones using a probabilistic random forest. Part I: Method and case studies"

_Weather and Climate Dynamics, 2022_

## Referee Comment (RC1)

**Review of "Objective Identification of high wind features within extratropical cyclones using a probabilistic random forest (RAMEFI). Part 1: Method and illustrative case studies" by L. Eisenstein, B. Schulz, G. A. Qadir, J. G. Pinto and P. Knippertz.**

This study aims to objectively identify different mesoscale wind features associated with extra-tropical cyclones by applying a probabilistic random forest approach. Overall, the topic is very interesting, the method novel, and very promising results are presented. My three main issues are (1) the structure of the paper needs to be improved and streamlined, (2) there are many terms concerning the random forest and its verification that a typical reader of WCD will not be able to understand and (3) there are notable limitations of this method that should be highlighted more clearly. These issues, and more minor points, are detailed below.

**Major Comments**

1. **The structure of the manuscript could be improved.** I appreciate that this is not a simple manuscript to organise, but I think with some careful re-structuring the manuscript would become shorter and much easier to read.

   (a) Currently there is quite a bit of repetition of the same or very similar topics. One main cause of this is the large amount of overlap between section 2 (theoretical description of the wind features) and section 4.1 (how they were labelled). I think if these could be combined, it would be much easier for a reader. Some specific examples of repetition include: lines 204-210, also the text in lines 196-199 is repeating information that has already been stated, Lines 259 -260.

   (b) It is not clear exactly what threshold / criteria were used to manually label the features. Was a certain set of "rules" created for each variable and each feature or was it done purely subjectively? In lines 315 – 318 it is mentioned some of the characteristics a WJ requires. It is also said that "signs for a CJ as missing" but it is not clear what these signs are. I think my confusing comes from the structure of the manuscript (see point above) with the "theoretical" characteristics described in section 2 but then the characteristics / values of the actual labelled features are loosely described in section 4 and 5. I strongly encourage the authors to reconsider the structure and combine the theoretical basis and what was actually used in the same section.

   (c) Methods are distributed throughout the manuscript breaking up the flow for a reader and making it hard to find the method details if a reader wants to go back and check. Furthermore, some details from the appendices should be included in the main text. Section 4 is headed "Method" but it needs to be better structured and expanded to include extra details / extra subsections of all of the methods used to develop "*the method*".

      1. First, a clear subsection of how the 12 cases were selected needs to be included somewhere as currently it is easy for a reader to miss. For this, some basic details of the SSI should be included in the main text, not in the appendix.

      2. All of section 6.1 is describing a method. It interrupts the flow of the results. Could this be moved to the "methods" section and revised to make it more accessible for a typical WCD reader (see below)?

3. Details of why the threshold of 80% of the 98$^{th}$ percentile is used should be more clearly stated. Currently, this is first stated on line 201 but it should be earlier. Furthermore, this should not be under a subheading "subjective labelling using an interactive tool" as it is really part of the method.

(d) In a few places in the text, future sections of the manuscript are referred to which is somewhat distracting for a reader but may also indicate that topics are not in the most optimal order (e.g. line 123, line 212, line 262, line 290, line 325, line 350, line 382, line 399).

2. **Limitations of the method should be clearly highlighted and considered more.**

(a) The requirement of having either observations or model data at 1 hour temporal resolution is a limitation of this study and potentially will limit how this method / software can be applied in the future e.g. data from CMIP models is generally not available at 1 hour time resolution and nor are observations from other parts of the world. This limitation should be stressed more clearly – it is briefly noted in section 3, where it is stated that few stations report precipitation with 1 hour resolution. Furthermore, the justification for using changes over 1 hour should be motivated and justified in the methods / data section.

(b) In Line 416 it is stated "The RF should only be applied to regions affected by extra-tropical cyclone". Given the text that comes before this, this seems a little inaccurate. Is it not more accurate to state that the RF should only be applied in the region where the training was performed (over Europe and over land)? This also raises concerns about how valid this method would be in other areas of world. A specific example is that extra-tropical cyclones certainly affect mountainous regions (e.g. Norway, Iceland) so is it likely that the RF method will not work in those area which certainly are commonly affected by extra-tropical cyclones? At a minimum some text should be added to considered how accurate the method will work over ocean and in other parts of the world e.g. at the start of the storm track, in mountainous areas.

(c) Subjective choices. Although the method is objective, the training and some threshold are subjective. It should be considered how sensitive the results are to the subjective training. It should also be considered how the results depend on the threshold used for "strong wind features" e.g. 80% of the 98$^{th}$ percentile? For examples, is it a limitation of this method that the same threshold is used for all wind features even though some are well known to be stronger than others? For example, a warm jet is likely to have weaker winds than a cold jet so does this mean fewer warm jet features are identified, or that more extreme WJs are identified that CJs.

(d) Are computation costs another limitation that should be more clearly noted. On line 231: "For computational reasons..." This needs to be more clearly explained. Does this mean that the method cannot work on a very large data set? It would be nice if some indication of the computing cost of this probabilistic method could be included in the manuscript.

3. **Technical terms are hard for a meteorologist / typical WCD reader to understand.** I think overall it is great that new methods are being used in the field of synoptic-dynamic

meteorology but not all readers of WCD will be familiar with these methods and the terminology that goes along with them (myself included). For these reason, special care is needs to ensure terms are clearly explained and also put into context of the data / synoptic terms used. It would also be clearer to explain some of these terms (particularly those about the evaluation of the forecasts) in the methods section rather than in the middle of the results (see # 1 above).

(a) Section 4.2 / machine learning terms. As a meteorologist I found this section very hard to understand and I am probably quite a typical reader of WCD. There were quite a few terms in this section that I do not understand (order criteria, predictor space, target variable). I feel that for this journal these terms need to be better described. Potentially some schematic diagram describing the probabilistic random forest could be included to make it easier for a reader to understand.

(b) Section 6.1 / terminology. Again, in this section there were quite a few terms that I did not fully understand. For example, "multi-class probability forecast", "binary predictions", "COPR-approach" (the abbreviation should be explained). For a reader it would help if these terms could be out into content with the case here. e.g. I think for "multi-class" class = the different wind features.

(c) Section 6.2 / terminology. More examples of terms that are not easy for a meteorologist to understand: "For each fold of the cross-validation scheme...",  "binary probabilistic forecasts"

**Minor Comments**

1. The title is very long. I don't think it is necessary to have the abbreviation "RAMEFI" in the title. "illustrative" could also be removed.

2. The synoptic meteorology terminology that is used is not clearly explained. Specifically, it is not clear whether the "short warm jet" is exactly the same as a warm conveyor belt (WCB), and whether the "short cold jet" is the same as a cold conveyor belt (CCB). How these terms relate to the commonly used terms (WCB, CCB) should be clearly defined in the introduction. Potentially also considered revising the subheadings of 2.1 and 2.2 as it is unclear why "conveyor belt" is in parenthesis.

3. Introduction – first 2 paragraphs. Here the Norwegian cyclone model and the Shapiro Keyser cyclone model are introduced. It would be beneficial to include a more in-depth discussion of how all of the wind features typically different between the two types of cyclone models – currently the only reason to introduce these different cyclone models appears to be to state that sting jets only occur in Shapiro-Keyser cyclones. This could be included in section 2 (or a revised combined section 2, section 4.1) rather than in the introduction.

4. Related to minor comment #3 - Lines 33 – 37 and Figure 1. This could be incorporated in section 2. Or at least Figure 1 should be referred to much more from the text in section 2 as currently Figure 1 is not referred to / utilised to its full extent.

5. Line 42. Can some additional details about the type of post-processing Pantillon et al (2021) used be added here?

6. After reading the introduction, I was wondering if any other studies had applied the probabilistic random forest method to meteorological data sets as its not clear from the introduction. If previous studies have used this method, it should be stated.

7. The discussion of the warm conveyor belt in section 2.1 does not consider the cyclonic and anticyclonic branches which are shown in Figure 1. These features should be briefly noted since they are shown in Figure 1.

8. Line 92. I disagree with the statement that there is little precipitation in the warm sector. The warm conveyor belt is often the cause of a significant proportion of precipitation in extra-tropical cyclones.

9. Line 108. Something is missing in the text here "...stays close to the ground 850 hPa..".

10. Section 2.3 / Sting jets. Please give some indication of the spatial and temporal scale of a sting jet.

11. Line 154. The computation of the potential temperature is unclear as is the reason why potential temperature is used over temperature. If mean sea level pressure (p) is used, and a reference pressure of 1000-hPa is used to compute the potential temperature, then the effect of station altitude is not removed – which was one reason I assumed potential temperature was used over temperature. Can both the method to compute potential temperature and the reason for doing so be more clearly explained in the manuscript as I think I have misunderstood something here.

12. Line 165. The justification for normalising the wind speed by its $98^{th}$ percentile is not clear here – is it to remove seasonal / diurnal cycle or to focus on high wind events?  I think it is the former and that the 0.8 threshold on v/v98 mentioned later is to ensure only high wind events are considered.

13. Line 170. Can the Euro Cordex domain be shown on a map or at least the longitudes / latitudes it covers added to the text?

14. Section 3.2. Can the vertical levels of the variables taken from COSMO-REA6 be added to the text?

15. Lines 290. Is this text really needed?

16. Line 293. (int. ) - this is not clear. Maybe it would be better to write also known elsewhere as Eleanor?

17. Line 322. Suggest to add "training labels" and "predictions" in parenthesis after the reference to Figs 3d and 4b.

18. Line 338. I don't understand what is meant by "the edges of the boundary or cyclone". Please revise and clarify this sentence.

19. Lines 342, 344. The terminology is a little inconsistent here and thus potentially confusing. In line 342, the comparison of station observations to gridded forecasts are discussed whereas in line 344 it is the comparison between  station observations and reanalysis-based results. If this is the same thing, please be consistent with the terminology. (forecasts made me think more of operational weather forecasts than the forecasts from RAMEFI)

20. Section 5.3 / comparison of Figure 5 to Figure 4b. Here the biggest difference is that in figure 5 the different wind features / different colours join up with each other (not many white grid points) whereas in Figure 4b there are gaps. Why is this?

21. Line 373. What is the "climatological benchmark"? If this is some average of the 12 cases, then I think using the term "climatological" is misleading.

22. Line 430. Again there is a term used that I do not understand – please explain what the "BS permutation importance" is.

**Figures and Tables**

1. Figure 1: The long cold front shown in this schematic is not typical of what a Shapiro-Keyser cyclone looks like. Could this front be shortened? Would figure 1 also have a panel b showing a Norwegain cyclone model type of cyclone as well?

2. Table 1. Can units be added to this table?

3. Figure 2. How are the cyclone tracks shown in this figure produced? e.g. from an objective tracking algorithm or subjectively? Also Eberherd does not seem to have any markers indicating MSLP or are they just too small to see?

4. Figure 3d. Are the labels only for stations where $v > 0.8$? This does not seem to be consistent with the contours e.g. there are labelled features outside of the dotted contour.

5. Figure 3 caption "set labels" in the caption is not a very clear term. Is "training labels" clearer?

6. Figure 4. Could the mean sea level pressure field be added to these panels to give a reader some synoptic context for where the different wind features are predicted and to get a rough idea of what the different cyclones looked like.

7. Figure 6: What does CEP stand for? Please clarify what "via consistency bands (under the assumption of calibration" means. These details should be in the text rather than just in the caption. Also for figure 6, are all 12 storms included? This information should be added to the caption.

8. Figure 7. Please define CEP.

9. Figures 8, 9 and 10. the delta symbol does not seem to be displaced correctly.

10. Figure 10. Please add the units for each of the predictor variables.

---

## Referee Comment (RC2)

Review – Ambrogio Volonté

**Objective identification of high-wind features within extratropical cyclones using a probabilistic random forest (RAMEFI). Part I: Method and illustrative case studies**

This manuscript contains a very interesting study, illustrating an objective approach at the identification of strong wind features in extratropical cyclones. The manuscript is well-written and insightful and the results shown are indeed promising. Novel methods are used, sometimes beyond the typical expertise of WCD readers (and reviewers!). On this, I would recommend addressing the main comments made by the first reviewer. My concerns echo those three key points and therefore I'm not repeating them here. I add below some other comments, all generally minor. I would be happy to see this manuscript accepted for publication once all these points are addressed.

**Comments:**

Abstract:

- Line 9, "of spatial dependencies and gradients": this is quite vague and not really clear.

Introduction:

Line 25, "belong to": I would write "can produce some of" or something similar.

Line 34: I would write "or, in short, the warm jet (WJ). Same point applies to the CJ.

Characteristics of high-wind features:

- Line 92, "little or no precipitation": can you provide a reference for this statement? Warm conveyor belts are, as you write, airstreams ascending in the warm sector of the cyclone, and are associated with vigorous moist processes (condensation, …). Therefore, the reader would be surprised to hear that little or no precipitation is associated with them.

Line 109, "ground 850 hPa": missing word here, perhaps "below 850 hPa"?

Section 2.5: I find feature naming not totally consistent here. If dry intrusion and post-cold-frontal convection are both subsets of CS, why does Figure 1 display CS + pCFC in the cold sector? Shouldn't it be DI + pCFC (= CS) ?

Data:

Line 153: "wind speed at 10m" (same applies to wind direction)

Line 153: I'm being pedantic here, but could you replace "RR" with "R", given that it's precipitation amount and not rain rate? (or if it's actually rain rate, please state it)

Method:

Line 218, "Section 4": this line is already in Section 4, so I guess this should be referring to a different section.

Line 219: I would argue that wind speeds are not "enhanced" by a strong pressure gradient, as this is what causes winds to be strong (at meso-synoptic scale) in the first place! In my view, factors enhancing wind speeds are those not accounted for by the (gradient-wind-) balanced flow, such as convective downdraughts for CFC and symmetric instability for SJ.

Table 2: could you include the height of max wind location? (and if it's > 800 m, as I think Zugspitze is, add the max winds below 800m?)

Figure 2: could you make the colour progression more intuitive? (e.g., using green and dark green in consecutive years, instead of having red in between)

Lines 225-226: could you mention the features (after NF) according to the number of points, in decreasing order? I think it would make for an easier reading of the sentence.

Line 228: I'm not sure I understand the rationale of merging CJ and SJ points. I would understand doing this if you weren't able to separate them, but you have just listed them separately, so I don't get why you then decide to put them back together. Is this because you think RAMEFI wouldn't be able to separate them? If so, state it explicitly.

Line 249-250: Yes, I think this is really important

Line 267: "Appendix A1".

Line 286: "are provided in Appendix A2 (practical implementation ) and B1 (mathematical formulation)".  (Do you reference to A3 anywhere? I couldn't find where)

Illustrative case study:

Figure 3: I'm not sure I'm reading it correctly, as it seems to me that also locations where $v <$ 0.8 (i.e., outside that dotted contours) are included in panel d. Also, I can't see any solid contours. Could you clarify?

Line 349: Where would a "hook-shaped structure of the winds" be considered in your algorithm?

Statistical evaluation:

Line 374: "in Appendix B2".

Lines 366-367, "Further, […] or 1". Could you rephrase this sentence? It is not very clear to me.

Discussion:

Figure 7 and all-pairs approach: Does the order of features matter? In other words, is (0 = CJ, 1 = CS) equal to (0 = CS, 1 = CJ). If that's not the case, the missing panels should be included and discussed.

Lines 476-478: Wouldn't high winds related to a strong synoptic-scale pressure gradient (which to me just means a deeper, more intense cyclone) still fall in the same categories (features) but with higher wind speed values than a shallower cyclone? Or are you implying that deeper cyclones have on average a different structure? Please clarify this.

Section 7.2: Still on the same point, but I think this is crucial for the understanding. I don't think the expression "the highest wind speeds are enhanced by a strong background pressure gradient" is physically correct (see my "Line 219" comment). I agree that with a deeper cyclone, more locations can record strong winds, but I would still assume that even if they're not directly related to CJ or WJ, they would still be located either in the warm or cold sector (and thus fall in the CS category in the latter case). Could it be that you need a WS category?

Lines 498-499: please explain the difference between front and convergence line here, to improve clarity.

Lines 539-540: One could argue that this shows that the set of surface parameters used to train the RF is broad enough to allow it to include both SJ and CJ in the "CJ" category. Are you ruling out that separating SJ from CJ in the observation would lead to RF correctly identifying them individually? If so, on what basis?

Sections 7.2 and 7.4: My understanding is that using normalised values (of pressure, theta, etc…) you can identify features even in cases departing from "your climatology" (i.e., the mean values in the 12 cases selected). However, in these sections, you show that "anomalous" cases can be a challenge for RAMEFI. Do you expect this issue to be partially or totally solved when RAMEFI is trained on more data? Is this explored in the Part II paper?

Conclusions:

Lines 574-575: What surface values do you think could best distinguish CJ from CS (apart from p)? Maybe spatial temperature/theta gradients could be useful? Otherwise, the most obvious to me would be cloud cover, but I'm not sure how many stations would have radiation/sunshine measurements.

Lines 588-589: This is indeed an encouraging result, but this sentence is probably too bold, given you've just looked at 12 cases in observations vs 1 reanalysis dataset. Maybe you could replace "should" with "could"?

---

## Author Comment (AC1)

**Review of "Objective Identification of high wind features within extratropical cyclones using a probabilistic random forest (RAMEFI). Part 1: Method and illustrative case studies" by L. Eisenstein, B. Schulz, G. A. Qadir, J. G. Pinto and P. Knippertz.**
**10.5194/wcd-2022-29**

This study aims to objectively identify different mesoscale wind features associated with extra-tropical cyclones by applying a probabilistic random forest approach. Overall, the topic is very interesting, the method novel, and very promising results are presented. My three main issues are (1) the structure of the paper needs to be improved and streamlined, (2) there are many terms concerning the random forest and its verification that a typical reader of WCD will not be able to understand and (3) there are notable limitations of this method that should be highlighted more clearly. These issues, and more minor points, are detailed below.

We thank you for your constructive comments, which helped to improve our paper. We revised the mathematical parts of the manuscript to better explain our method and addressed the mentioned limitations. We understand that the structure might not be optimal and made several smaller changes to it and removed repetitions as suggested, while keeping the overall structure.

Below are the responses to your individual comments in blue. Text changes are included in italics when suitable. Line numbers correspond to the revised manuscript.

**Major Comments**

1. **The structure of the manuscript could be improved.** I appreciate that this is not a simple manuscript to organise, but I think with some careful re-structuring the manuscript would become shorter and much easier to read.

   (a) Currently there is quite a bit of repetition of the same or very similar topics. One main cause of this is the large amount of overlap between section 2 (theoretical description of the wind features) and section 4.1 (how they were labelled). I think if these could be combined, it would be much easier for a reader. Some specific examples of repetition include: lines 204-210, also the text in lines 196-199 is repeating information that has already been stated, Lines 259 -260.

   We moved the statements about spatial independence from the different subsections to Section 4 to avoid repetition (former lines 196-199 and 259-260, now lines 239-241).

   Further, we shortened the paragraph about how the features are labelled in Section 4.1 to reduce repetition and tried to clarify the point raised in your comment (b) below. However, we decided not to fully combine Sections 2 and 4.1, as Section 2 is discussing scientific literature and should stand for itself, while Section 4.1 is describing our method, hence also how we distinguish the features.

   (b) It is not clear exactly what threshold / criteria were used to manually label the features. Was a certain set of "rules" created for each variable and each feature or was it done purely subjectively? In lines 315 – 318 it is mentioned some of the characteristics a WJ requires. It is also said that "signs for a CJ as missing" but it is not clear what these signs are. I think my confusing comes from the structure of the manuscript (see point above) with the "theoretical" characteristics described in section 2 but then the characteristics / values of the actual labelled features are loosely described in section 4 and 5. I strongly encourage the authors to reconsider the structure and combine the theoretical basis and what was actually used in the same section.

We tried to clarify the main guidelines for labelling in Section 4.1 without repetition of Section 2:

"*The guiding principles for the labelling were extracted from the scientific literature (cf. Sect. 2) and are mainly based on the location relative to the cold front and cyclone core. […]. In our surface parameters, a cold front is then mostly identified through the characteristic change of the sign of Δp. It is labelled CFC, if a larger area of precipitation along it is observed, while high winds ahead of the front within the warm sector are labelled WJ. The CJ is mostly detected through its hook-shaped wind footprint at the tip of a wrapped-around occlusion or bent-back front as well as through its proximity to the cyclone centre. An SJ is labelled […].*" (lines 255-261)

We further revised Section 5.2 and clarified the missing signs of a CJ: "*As the occluded front is not wrapped around the core yet (cf. Fig. D4), implying that a CJ is not yet occurring at this point in time, […].*" (lines 366-367)

*(c)* Methods are distributed throughout the manuscript breaking up the flow for a reader and making it hard to find the method details if a reader wants to go back and check. Furthermore, some details from the appendices should be included in the main text. Section 4 is headed "Method" but it needs to be better structured and expanded to include extra details / extra subsections of all of the methods used to develop "*the method".*

We renamed Section 4 "*RAMEFI*" to introduce our own developed method while including used methods developed by others in Section 3 – renamed "*Data and methods*".

1. First, a clear subsection of how the 12 cases were selected needs to be included somewhere as currently it is easy for a reader to miss. For this, some basic details of the SSI should be included in the main text, not in the appendix.

   We moved the paragraph explaining the selection to a subsection in Section 3 (Section 3.3; line 200 ff.). Moreover, we added the concept of the SSI – while referring to the Appendix for further information – to Section 3.1. when first mentioned (see major comment 2c; lines 182-184).

2. All of section 6.1 is describing a method. It interrupts the flow of the results. Could this be moved to the "methods" section and revised to make it more accessible for a typical WCD reader (see below)?

   We revised and moved this subsection to Section 3 (Section 3.4; line 210 ff.)

3. Details of why the threshold of 80% of the 98th percentile is used should be more clearly stated. Currently, this is first stated on line 201 but it should be earlier. Furthermore, this should not be under a subheading "subjective labelling using an interactive tool" as it is really part of the method.

   We moved this to the first paragraph of Section 4 (lines 236-238). The reasoning behind the 98th percentile has already been added to Section 3.1 (see response to major comment 2c; lines 181-182).

(d) In a few places in the text, future sections of the manuscript are referred to which is somewhat distracting for a reader but may also indicate that topics are not in the most optimal order (e.g. line 123, line 212, line 262, line 290, line 325, line 350, line 382, line 399).

We removed the references in line 123, line 262, line 290, and line 325. Few others we

decided to keep, as we feel that they are useful for the readers.

2. **Limitations of the method should be clearly highlighted and considered more.**

   (a) The requirement of having either observations or model data at 1 hour temporal resolution is a limitation of this study and potentially will limit how this method / software can be applied in the future e.g. data from CMIP models is generally not available at 1 hour time resolution and nor are observations from other parts of the world. This limitation should be stressed more clearly – it is briefly noted in section 3, where it is stated that few stations report precipitation with 1 hour resolution. Furthermore, the justification for using changes over 1 hour should be motivated and justified in the methods / data section.

   Looking at features like CFC, a temporal resolution of less than 1 hour is too coarse, as it is usually a narrow feature affecting an area only for a short time. Furthermore, the trend in our community is going towards the provision of hourly data anyway, especially for surface parameters, and this work is motivated by the possible improvement of forecasts, which are usually available hourly.

   We added a short sentence on this in Section 4: "*To capture usually narrow and fast-moving features such as CFC, RAMEFI requires hourly data.*" (lines 238-239)

   (b) In Line 416 it is stated "The RF should only be applied to regions affected by extra-tropical cyclone". Given the text that comes before this, this seems a little inaccurate. Is it not more accurate to state that the RF should only be applied in the region where the training was performed (over Europe and over land)? This also raises concerns about how valid this method would be in other areas of world. A specific example is that extra-tropical cyclones certainly affect mountainous regions (e.g. Norway, Iceland) so is it likely that the RF method will not work in those area which certainly are commonly affected by extra-tropical cyclones? At a minimum some text should be added to considered how accurate the method will work over ocean and in other parts of the world e.g. at the start of the storm track, in mountainous areas.

   The RF should generally be applied with caution in regions it was not trained on. However, we don't include location-specific parameters and normalise temperature and wind with the local climatology, so we believe that the RF could be applied to other regions where extratropical cyclones occur and we want to motivate further studies testing this.

   We included further discussion in the end of Section 4.2.: "*Due to normalising θ and v, the trained RF is fairly independent of location-specific information, such that it can hopefully be applied successfully to other midlatitude regions around the world affected by extratropical cyclones. However, before doing that we recommend a thorough sanity check, particularly when using it over the ocean and mountainous regions.*" (lines 318-320)

   Moreover, we added the following sentence to the conclusions: "*Although applying RAMEFI over regions other than that used in the training has not been examined yet, relying on location-independent predictors suggests that it should be possible with no or only little modification.*" (lines 625-627)

   (c) Subjective choices. Although the method is objective, the training and some thresholds are subjective. It should be considered how sensitive the results are to the subjective training. It should also be considered how the results depend on the threshold used for

"strong wind features" e.g. 80% of the 98[th] percentile? For examples, is it a limitation of this method that the same threshold is used for all wind features even though some are well known to be stronger than others? For example, a warm jet is likely to have weaker winds than a cold jet so does this mean fewer warm jet features are identified, or that more extreme WJs are identified that CJs.

We aimed to make RAMEFI as objective as possible and thus included only one threshold – 80% of the 98[th] percentile. The 98[th] percentile is used, as it is a standard quantity indicating exceptionally high winds, e.g., for the calculation of the SSI. The idea behind our method is to find the meteorological cause of a wind speed above a certain high but not extreme level, which is why we decided to use the same relative threshold for all features. Different thresholds for the different features would mean adding a subjective criterion, while the random forest in principle can learn how strong wind speeds typically are for given feature. However, we saw that the normalised wind speed is not particularly important for the decision between the different features.

We added a sentence on why we use the 98[th] percentile in Section 3.1 when we introduce the normalised wind speed: "*The 98[th] percentile is used in analogy to standard high-wind quantities such as the Storm Severity Index (SSI), which is computed from stations where measured gusts exceed the local 98[th] percentile and provides an integral indication for the strength of the cyclone and the associated potential damage (see Appendix C for details).*" (lines 181-184)

Furthermore, we clarified the usage in Section 4: "*Our new method RAMEFI focuses on strong but not exceptionally high wind speeds. The latter are usually indicated by the 98[th] percentile. To obtain a sufficiently large storm area and to base that on a widely used reference, we decided to include stations reaching 80% of their 98[th] percentile, i.e., v ≥ 0.8.*" (lines 236-238)

(d) Are computation costs another limitation that should be more clearly noted. On line 231: "For computational reasons..." This needs to be more clearly explained. Does this mean that the method cannot work on a very large data set? It would be nice if some indication of the computing cost of this probabilistic method could be included in the manuscript.

The computational costs have to be considered using the labelling tool and when training the RF. Once trained, the RF can be used also on large data sets.

In Section 4.1. we clarified this: "*For computational reasons, i.e., as labels are set for every grid point rather than an area, we downsampled the COSMO grid […].*" (lines 276-277)

3. **Technical terms are hard for a meteorologist / typical WCD reader to understand.** I think overall it is great that new methods are being used in the field of synoptic-dynamic meteorology but not all readers of WCD will be familiar with these methods and the terminology that goes along with them (myself included). For these reason, special care is needs to ensure terms are clearly explained and also put into context of the data / synoptic terms used. It would also be clearer to explain some of these terms (particularly those about the evaluation of the forecasts) in the methods section rather than in the middle of the results(see # 1 above).

(a) Section 4.2 / machine learning terms. As a meteorologist I found this section very hardto understand and I am probably quite a typical reader of WCD. There were quite a few terms in this section that I do not understand (order criteria, predictor space, target variable). I feel that for this journal these terms need to be better described. Potentially

some schematic diagram describing the probabilistic random forest could be included to make it easier for a reader to understand.

*Thank you for pointing this out. We tried to rephrase Sections 4.2 and 4.3 such that they are adequate for readers that are not familiar with machine learning terminology. The description of the decision trees was restructured to help the reader understand the method.*

(b) Section 6.1 / terminology. Again, in this section there were quite a few terms that I did not fully understand. For example, "multi-class probability forecast", "binary predictions", "COPR-approach" (the abbreviation should be explained). For a reader it would help if these terms could be out into content with the case here. e.g. I think for "multi-class" class = the different wind features.

*Thank you for pointing this out. Again, we rephrased the paragraph and directly related the methodology to the application, i.e., the identification of the different wind features. We included a short explanation of the CORP abbreviation.*

(c) Section 6.2 / terminology. More examples of terms that are not easy for a meteorologist to understand: "For each fold of the cross-validation scheme...", "binary probabilistic forecasts"

*In line with (b) and (c), we rephrased the paragraph.*

**Minor Comments**

1. The title is very long. I don't think it is necessary to have the abbreviation "RAMEFI" in the title. "illustrative" could also be removed.

   *We agree and this has been changed accordingly. Additionally, we removed "objective" as we think that the use of a probabilistic random forest already covers that.*

2. The synoptic meteorology terminology that is used is not clearly explained. Specifically, it is not clear whether the "short warm jet" is exactly the same as a warm conveyor belt (WCB), and whether the "short cold jet" is the same as a cold conveyor belt (CCB). How these terms relate to the commonly used terms (WCB, CCB) should be clearly defined in the introduction. Potentially also considered revising the subheadings of 2.1 and 2.2 as it is unclear why "conveyor belt" is in parenthesis.
   *We apologise for the confusion. We did not mean to introduce a "short warm jet" but rather introduce a shorter term for "warm conveyor belt jet" (analogously for cold jet). That is also why we have "conveyor belt" in the parenthesis. We decided to use WJ and CJ to clearly distinguish the warm/cold jet from the warm/cold conveyor belt.*
   *As suggested by Reviewer 2, we changed the sentence to "[…] the warm conveyor belt jet or, in short, warm jet (WJ), the cold conveyor belt jet or, in short, cold jet (CJ), […]." (line 34)*
   *Furthermore, we changed the subheadings to "Warm jet" and "Cold jet".*
   *We decided to clarify the relation and differences between WCB/CCB and WJ/CJ in Section 2 (see minor comments #7 and #8).*

3. Introduction – first 2 paragraphs. Here the Norwegian cyclone model and the Shapiro Keyser cyclone model are introduced. It would be beneficial to include a more in-depth discussion of how all of the wind features typically different between the two types of cyclone models – currently the only reason to introduce these different cyclone models appears to be to state that sting jets only occur in Shapiro-Keyser cyclones. This could be included in section 2 (or a revised combined section 2, section 4.1) rather than in the introduction.

As most features are expected to have the same characteristics in both models, we only included the following sentence (also regarding your comment to Fig. 1): "*Figure 1 exemplarily shows a typical SKC; differences to an NC are discussed in the following.*" (line 86)

This is followed by the statements that SJ only occur in SKC and that CFC is less common in SKC.

4. Related to minor comment #3 - Lines 33 – 37 and Figure 1. This could be incorporated in section 2. Or at least Figure 1 should be referred to much more from the text in section 2 as currently Figure 1 is not referred to / utilised to its full extent.

Indeed, Figure 1 was not discussed a lot in Section 2. We included it in the new discussion of WCB-WJ (see minor comments #2, #7, #8) and in further instances. However, we believe that the referenced lines and first occurrence of Figure 1 should remain in Section 1 for a first introduction, as these high-wind features are the focus of this work. We added a reference to Section 2 for more details on the features.

5. Line 42. Can some additional details about the type of post-processing Pantillon et al (2021)used be added here?

Thank you for pointing this out. Indeed, we did not specify the type of post-processing used in Pantillon et al. (2018) and would have confused a reader by mentioning neural network-based methods in the previous sentence, which are not used by Pantillon et al. (2018). Rather a much simpler approach (that was also included in Schulz and Lerch (2022) and significantly improved reliability and accuracy) was used. We clarified this by rephrasing both sentences.

We added this information to the introduction: "*However, Pantillon et al. (2018), who applied one of the classical statistical methods to ensemble forecasts of wind gusts […].*" (lines 44-45)

6. After reading the introduction, I was wondering if any other studies had applied the probabilistic random forest method to meteorological data sets as its not clear from the introduction. If previous studies have used this method, it should be stated.

Thank you for pointing this out. Indeed, we did not refer to any other studies that had applied probabilistic RFs to meteorological data. Therefore, we included three references of applications of RFs in the context of weather forecasting in Section 4.2: "In a meteorological context, probabilistic RFs have already been applied to predict damaging convective winds (Lagerquist et al., 2017) and severe weather (Hill et al., 2020), but also in a general form for a wide range of applications such as ensemble post-processing of surface temperature and wind speed (Taillardat et al., 2016)." (lines 294-296)

To the best of our knowledge, RFs have not been applied in the context of European winter storms and/or the prediction of different wind features. Currently, RFs are widely used and thus there may be other relevant references that we are not aware of.

7. The discussion of the warm conveyor belt in section 2.1 does not consider the cyclonic and anticyclonic branches which are shown in Figure 1. These features should be briefly noted since they are shown in Figure 1.

We apologise for this. As our focus was on the WJ and not the WCB, we indeed neglected these characteristics. We included the following information:

"*During the ascent, the WCB splits into a cyclonic and anticyclonic branch as seen by the red tubes in Fig. 1. While the cyclonic part forms the cloud head and usually causes heavy*

*precipitation along a narrow region, the anticyclonic part rises above the warm front and brings more moderate precipitation over a wider area. Overall, the WCB is the main cause for long-lasting precipitation (Catto, 2016). Furthermore, the WCB can be the cause of strong convection along the cold front (Hewson and Neu, 2015).*

*Contrary to Hewson and Neu (2015), we define the WJ as the region ahead of the cold front and its convection, hence ahead of the CFC feature (cf. Sect. 2.4), as displayed by the red shaded ellipse in Fig.1.*" (lines 95-101)

8. Line 92. I disagree with the statement that there is little precipitation in the warm sector. The warm conveyor belt is often the cause of a significant proportion of precipitation in extra- tropical cyclones.

We agree that the WCB is responsible for most of the precipitation associated with a cyclone, however, it occurs along the fronts and not within the warm sector. Contrary to Hewson and Neu (2015), we split the feature into WJ ahead of the (rainy) cold front and CFC along the front. A short clarification has been included in the revision of Section 2.1 together with minor comment #7.

9. Line 108. Something is missing in the text here "...stays close to the ground 850 hPa..".

Thank you for the notice, we changed the text to "*[…] stays close to the ground, i.e., below 850 hPa […].*" (line 120)

10. Section 2.3 / Sting jets. Please give some indication of the spatial and temporal scale of a sting jet.

We added the following sentence to the section: "*Hewson and Neu (2015) suggest an average surface footprint of less than 100 km in width and up to 800 km in length. SJs usually last just a few hours but can be active up to 12 h in extreme cases.*" (lines 129-131)

11. Line 154. The computation of the potential temperature is unclear as is the reason why potential temperature is used over temperature. If mean sea level pressure (p) is used, and a reference pressure of 1000-hPa is used to compute the potential temperature, then the effect of station altitude is not removed – which was one reason I assumed potential temperature was used over temperature. Can both the method to compute potential temperature and the reason for doing so be more clearly explained in the manuscript as I think I have misunderstood something here.

We calculated the surface pressure using the barometric height formula in observations to then use it for the calculation of the potential temperature. For COSMO-REA6 the surface pressure was available in the data set.

For clarification we changed the text to: "*Using T and p, we further compute the surface pressure using the barometric height formula to then calculate the potential temperature.*" (lines 169-170)

12. Line 165. The justification for normalising the wind speed by its 98[th] percentile is not clear here – is it to remove seasonal / diurnal cycle or to focus on high wind events? I think it is the former and that the 0.8 threshold on v/v98 mentioned later is to ensure only high wind events are considered.

On the one hand, normalising the wind speed by the 98[th] percentile is to remove the seasonal / diurnal cycle, but also location-specific characteristics, e.g., higher wind speeds caused by the exposed location of a station. On the other hand, both the normalisation and using the 0.8 threshold are to focus on high but not extreme wind events (see major comment 2c).

13. Line 170. Can the Euro Cordex domain be shown on a map or at least the longitudes / latitudes it covers added to the text?

As the focus is not on the Euro Cordex domain, but the dataset has been regridded, we decided to include the longitudes/latitudes of the new area instead: "*The data, originally on a rotated grid, are regridded to a latitude-longitude grid with a grid spacing of 0.0625°, i.e., roughly 7 km, for the area of -10°E to 30°E, 40°N to 65°N.*" (lines 198-199)

If parties are interested in the Euro Cordex domain, it can easily be found in the referred publication of Bollmeyer et al. (2015).

14. Section 3.2. Can the vertical levels of the variables taken from COSMO-REA6 be added to the text?

Although the COSMO-REA6 dataset includes 3D fields of some parameters, we only included the surface parameters, which are available for our observation dataset, so far.

We clarified this by adding "*The same surface parameters as mentioned in Sect. 3.1 are used*" before listing the parameters from the data set (lines 194-195).

15. Lines 290. Is this text really needed?
We removed the reference.

16. Line 293. (int.) - this is not clear. Maybe it would be better to write also known elsewhere as Eleanor?

This has been changed accordingly (line 344).

17. Line 322. Suggest to add "training labels" and "predictions" in parenthesis after the reference to Figs 3d and 4b.
Analogous to the change of the caption of Fig. 3 (see comment below) we added "*subjectively identified features*" and "*RF predictions*" (line 373).

18. Line 338. I don't understand what is meant by "the edges of the boundary or cyclone". Please revise and clarify this sentence.
We apologise for the confusion. This was changed to "*[…] at the peripheries of a cyclone or not connected to one […].*" (line 389)

19. Lines 342, 344. The terminology is a little inconsistent here and thus potentially confusing. In line 342, the comparison of station observations to gridded forecasts are discussed whereas in line 344 it is the comparison between station observations and reanalysis-based results. If this is the same thing, please be consistent with the terminology. (forecasts made me think more of operational weather forecasts than the forecasts from RAMEFI)

We apologize for the confusion here as well. We changed "forecasts" to "*data sets*" (line 393). RAMEFI enables us to compare observations with any hourly gridded data sets, while we use a reanalysis data set here.

20. Section 5.3 / comparison of Figure 5 to Figure 4b. Here the biggest difference is that in figure 5 the different wind features / different colours join up with each other (not many white grid points) whereas in Figure 4b there are gaps. Why is this?

The main reason is that in Figure 5 we have complete information on a dense regular grid, while the stations are distributed irregularly, do not measure all parameters at all times and have to be interpolated. Furthermore, the values at neighboring grid points in COSMO-REA6 are related to each other through the process of data assimilation in contrary to the more independent station observations.

We added the following sentence to the manuscript: "*Note that COSMO-REA6 provides*

*complete information on a dense regular grid in contrast to the irregularly distributed stations that have to be interpolated. This leads to more coherent areas here.*" (lines 397-398)

21. Line 373. What is the "climatological benchmark"? If this is some average of the 12 cases, then I think using the term "climatological" is misleading.

Indeed, the term climatological is misleading in this context. The benchmark prediction for one storm is given by the class frequencies of the wind features within the other 11 storms. We rephrased the corresponding formulations (line 413).

22. Line 430. Again there is a term used that I do not understand – please explain what the "BS permutation importance" is.

The BS permutation importance is a predictor importance technique introduced in Section 3.4 (former Section 6.1). Following major comment (3c), we rephrased this section, which also included the description of the BS permutation importance.

**Figures and Tables**

1. Figure 1: The long cold front shown in this schematic is not typical of what a Shapiro-Keyser cyclone looks like. Could this front be shortened? Would figure 1 also have a panel b showing a Norwegain cyclone model type of cyclone as well?

We would not say that a long cold front is not typical with a SKC. Many SKC have a rather straight, long, but weak cold front, e.g., storm Christian/St Judes Day storm (Browning et al., 2015; doi: 10.1002/qj.2581). Furthermore, the original schematic is taken from Clark and Gray (2018) and only adapted to include CFC and CS footprints. We think a second panel showing a Norwegian cyclone is not necessary, however, we added a comment on this as mentioned for minor comment #3.

2. Table 1. Can units be added to this table?

Yes, units have been added to the table.

3. Figure 2. How are the cyclone tracks shown in this figure produced? e.g. from an objective tracking algorithm or subjectively? Also Eberherd does not seem to have any markers indicating MSLP or are they just too small to see?

Here, we used a simple objective tracking algorithm searching for the pressure minimum and connecting it to following time steps. Indeed, for Eberhard (and also Fabienne) the markers are very small as the MSLP minimum was not particularly low, such that they can rarely be seen. We increased the size of the markers slightly to improve this.

4. Figure 3d. Are the labels only for stations where v > 0.8? This does not seem to be consistent with the contours e.g. there are labelled features outside of the dotted contour.
We apologise for the confusion. We used a simple interpolation approach to bring the observations onto a grid. However, this did not work very well. Now, as we did for the feature probabilities, we use Kriging to interpolate v and added a smoothing step to remove some noise, which improves things overall. However, some stations are still outside the contours, although v>0.8 was measured. Despite this, we feel that the changes are satisfactory now.

5. Figure 3 caption "set labels" in the caption is not a very clear term. Is "training labels" clearer?

As we use the labels also for the evaluation and not only for training, we decided to call them "*subjectively identified wind features*" instead.

6. Figure 4. Could the mean sea level pressure field be added to these panels to give a reader some synoptic context for where the different wind features are predicted and to get a rough idea of what the different cyclones looked like.

We included MSLP from the COSMO-REA6 reanalysis in Fig. 4, 5 and 11 as contours in light grey to avoid overloading the figures.

7. Figure 6: What does CEP stand for? Please clarify what "via consistency bands (under the assumption of calibration" means. These details should be in the text rather than just in the caption. Also for figure 6, are all 12 storms included? This information should be added to the caption.

When rephrasing Section 3.4 (former Section 6.1) for the major comment (3c), we included the definition of the CEP as well as information on the uncertainty quantification via consistency bands. For Figure 6 (and 7), all 12 storms are included, as mentioned in Appendix A3. To highlight this fact, we additionally provide this information in the caption of Figure 6 (and 7).

8. Figure 7. Please define CEP.

See previous comment.

9. Figures 8, 9 and 10. the delta symbol does not seem to be displaced correctly.

We tried hard but in the end were not able to make the Delta symbols align. We feel that this problem is rather cosmetic and does not obstruct the understanding of the figures' content.

10. Figure 10. Please add the units for each of the predictor variables.

Changed as suggested.

---

## Author Comment (AC2)

**Review – Ambrogio Volonté**
**Objective identification of high-wind features within extratropical cyclones using a probabilistic random forest (RAMEFI). Part I: Method and illustrative case studies**
**10.5194/wcd-2022-29**

This manuscript contains a very interesting study, illustrating an objective approach at the identification of strong wind features in extratropical cyclones. The manuscript is well-written and insightful and the results shown are indeed promising. Novel methods are used, sometimes beyond the typical expertise of WCD readers (and reviewers!). On this, I would recommend addressing the main comments made by the first reviewer. My concerns echo those three key points and therefore I'm not repeating them here. I add below some other comments, all generally minor. I would be happy to see this manuscript accepted for publication once all these points are addressed.

> Dear Ambrogio,
>
> We thank you for your valuable comments that helped to improve this manuscript. For the three key points made by reviewer 1 please see the corresponding reply. We further clarified the remaining open issues you made us aware of.
>
> Below are the responses to your individual comments in blue. Text changes are included in italics when suitable. Line numbers correspond to the revised manuscript.

Comments:
Abstract:
- Line 9, "of spatial dependencies and gradients": this is quite vague and not really clear.

> This was changed to "*[…] independent of local characteristics and horizontal gradients, […].*" (line 9)

Introduction:
- Line 25, "belong to": I would write "can produce some of" or something similar.

> This has been changed accordingly (line 25).

- Line 34: I would write "or, in short, the warm jet (WJ). Same point applies to the CJ.

> This has been changed accordingly (line 34).

Characteristics of high-wind features:

- Line 92, "little or no precipitation": can you provide a reference for this statement? Warm conveyor belts are, as you write, airstreams ascending in the warm sector of the cyclone, and are associated with vigorous moist processes (condensation, ...). Therefore, the reader would be surprised to hear that little or no precipitation is associated with them.

*We clarified the distinction between the WCB and our definition of WJ as also asked by Reviewer 1:*

*"During the ascent, the WCB splits into a cyclonic and anticyclonic branch as seen by the red tubes in Fig. 1. While the cyclonic part forms the cloud head and usually causes heavy precipitation along a narrow region, the anticyclonic part rises above the warm front and brings more moderate precipitation over a wider area. Overall, the WCB is the main cause for long-lasting precipitation (Catto, 2016). Furthermore, the WCB can be the cause of strong convection along the cold front (Hewson and Neu, 2015).*

*Contrary to Hewson and Neu (2015), we define the WJ as the region ahead of the cold front and its convection, hence ahead of the CFC feature (cf. Sect. 2.4), as displayed by the red shaded ellipse in Fig.1"* (lines 95-101)

- Line 109, "ground 850 hPa": missing word here, perhaps "below 850 hPa"?

*This has been changed to "[…] stays close to the ground, i.e., below 850hPa […]."* (line 120)

- Section 2.5: I find feature naming not totally consistent here. If dry intrusion and post-cold-frontal convection are both subsets of CS, why does Figure 1 display CS + pCFC in the cold sector? Shouldn't it be DI + pCFC (= CS) ?

*In an earlier version we included pCFC as a separate feature to see if its characteristics are closer to CS or CFC but it did not occur enough in observation data to derive meaningful statistics. We understand that it might be confusing in the current version and excluded it from Figure 1.*

Data:
- Line 153: "wind speed at 10m" (same applies to wind direction)

*This has been changed accordingly (line 168).*

- Line 153: I'm being pedantic here, but could you replace "RR" with "R", given that it's precipitation amount and not rain rate? (or if it's actually rain rate, please state it)

*We state that we have hourly data including precipitation amount, hence a rate.*

Method:
- Line 218, "Section 4": this line is already in Section 4, so I guess this should be referring to a different section.

*Thank you for making us aware of this. We removed the reference as we meant to refer to a former paragraph in the same subsection.*

- Line 219: I would argue that wind speeds are not "enhanced" by a strong pressure gradient, as this is what causes winds to be strong (at meso-synoptic scale) in the first place! In my

view, factors enhancing wind speeds are those not accounted for by the (gradient-wind-) balanced flow, such as convective downdraughts for CFC and symmetric instability for SJ.

> We agree and apologise for the misleading phrasing. We changed the sentence to "*[…], with an exceptional large pressure gradient leading to a stronger background wind field, such that […]*." (line 207)

- Table 2: could you include the height of max wind location? (and if it's > 800 m, as I think Zugspitze is, add the max winds below 800m?)

> We included the heights and have now two columns with maximum gusts – one above and one below 800m.

- Figure 2: could you make the colour progression more intuitive? (e.g., using green and dark green in consecutive years, instead of having red in between)

> Thank you for this advice. The order now matches a rainbow pattern.

- Lines 225-226: could you mention the features (after NF) according to the number of points, in decreasing order? I think it would make for an easier reading of the sentence.

> This has been changed accordingly (line 269).

- Line 228: I'm not sure I understand the rationale of merging CJ and SJ points. I would understand doing this if you weren't able to separate them, but you have just listed them separately, so I don't get why you then decide to put them back together. Is this because you think RAMEFI wouldn't be able to separate them? If so, state it explicitly.

> We labelled both SJ and CJ individually and trained an RF with these features separately to see if a distinction is possible with the used surface parameters alone. Unfortunately, but as expected, this was not successful, such that we included the SJ in the CJ category.

> We added the following for clarification: "*A first training with SJ and CJ as separate features showed that a clear distinction is not possible with the information at hand and that the SJ is mostly detected as CJ. Therefore, we decided to include it in the more frequent CJ feature […]*." (lines 271-273)

- Line 249-250: Yes, I think this is really important

- Line 267: "Appendix A1".

> This has been changed accordingly (line 317). Note that the order of the appendices changed.

- Line 286: "are provided in Appendix A2 (practical implementation) and B1 (mathematical formulation)". (Do you reference to A3 anywhere? I couldn't find where)

> This has been changed accordingly (lines 233-234). Thank you for pointing out that we did not refer to Appendix A3. We included a reference in Section 3.4 (former Section 6.1; line 337).

Illustrative case study:

- Figure 3: I'm not sure I'm reading it correctly, as it seems to me that also locations where v < 0.8 (i.e., outside that dotted contours) are included in panel d. Also, I can't see any solid contours. Could you clarify?

> We apologize for the confusion. We decided to include contours to indicate the regions of high winds. However, we used a simple interpolation approach to get the observations on a grid. Indeed, this did not work well enough. As we did for the feature probabilities, we now use Kriging to interpolate v and added a smoothing step to remove some noise. However, this leads to some stations still being outside the contours, although v>0.8 was measured. We hope that the changes are still satisfactory.

- Line 349: Where would a "hook-shaped structure of the winds" be considered in your algorithm?

> The hook-shaped structure is not considered in the algorithm. While we used this characteristic for the detection of a CJ, the RF seems to be able to detect it without. However, this might be one of the reasons, why the distinction between CJ and CS can appear difficult at times.

> We added a short note on this in Section 5.3: "*However, the occurrence of a hook-shaped structure cannot be accounted for in the spatially independent approach of RAMEFI making it difficult to distinguish these otherwise similar features.*" (lines 402-203)

> And further in Section 6.1: "*The main meteorological reason for this problem is the general similarity of the two features and that the hook-shaped structure, which is used for the subjective identification of a CJ, cannot be considered in the RF, such that […].*" (lines 432-434)

Statistical evaluation:

- Line 374: "in Appendix B2".

> This has been changed accordingly (line 337).

- Lines 366-367, "Further, [...] or 1". Could you rephrase this sentence? It is not very clear to me.

> Thank you for pointing this out. We rephrased the sentence by using the term "*confidence*" and not "certainty" (line 221). A prediction is said to be sharper, the more confident it is. In case of probabilities, a prediction is more confident if the predicted probability is closer to 0% or 100%, which are the most confident, i.e., the sharpest, predictions one can make.

Discussion:
- Figure 7 and all-pairs approach: Does the order of features matter? In other words, is (0 = CJ, 1 = CS) equal to (0 = CS, 1 = CJ). If that's not the case, the missing panels should be included and discussed.

> No, the order of the features does not matter, it only affects the orientation. If we choose (0 = CJ, 1 = CS), then a predicted probability of 70% corresponds to the occurrence of a

CS, and if we choose (0 = CS, 1 = CJ), it corresponds to the occurrence of a CJ. Hence, the two reliability diagrams of the two orderings are not identical but (some kind of) symmetric to each other and contain no additional information with respect to each other. Therefore, it is sufficient to consider only one of the two cases.

- Lines 476-478: Wouldn't high winds related to a strong synoptic-scale pressure gradient (which to me just means a deeper, more intense cyclone) still fall in the same categories (features) but with higher wind speed values than a shallower cyclone? Or are you implying that deeper cyclones have on average a different structure? Please clarify this.

Indeed, the same categories should apply for deeper cyclones, however, the distinction between the features is made more difficult and one cannot be certain if a feature really occurred or if the high winds are only caused by a rather unstructured background pressure gradient. We changed this to "*[…] sometimes high winds are mainly related to an exceptionally strong synoptic-scale pressure gradient […]*" (lines 512-513) and included more clarification in Section 7.2 (see next comment).

- Section 7.2: Still on the same point, but I think this is crucial for the understanding. I don't think the expression "the highest wind speeds are enhanced by a strong background pressure gradient" is physically correct (see my "Line 219" comment). I agree that with a deeper cyclone, more locations can record strong winds, but I would still assume that even if they're not directly related to CJ or WJ, they would still be located either in the warm or cold sector (and thus fall in the CS category in the latter case). Could it be that you need a WS category?

Again, we apologise for the misleading phrasing and confusion. We clarified the issue as also discussed in comments above in the first paragraph of Section 7.2.: "[…] unconnected to one of the four mesoscale wind features under study but can be enhanced by them. With an underlying strong wind field, the detection and distinction of the features might be more complicated." (lines 547-548)

- Lines 498-499: please explain the difference between front and convergence line here, to improve clarity.

We added the following sentence to the first paragraph of the subsection: "*While a cold front is associated with a second low pressure trough, a convergence line develops where two airflows collide and can occur independently of a cyclone.*" (lines 525-526)

- Lines 539-540: One could argue that this shows that the set of surface parameters used to train the RF is broad enough to allow it to include both SJ and CJ in the "CJ" category. Are you ruling out that separating SJ from CJ in the observation would lead to RF correctly identifying them individually? If so, on what basis?

As mentioned prior (see comment on line 228) we trained a RF with both features separately and added a comment on this in Section 4.1. Here, we want to show that the RF detects the SJ within the CJ feature as argued.

- Sections 7.2 and 7.4: My understanding is that using normalised values (of pressure, theta, etc...) you can identify features even in cases departing from "your climatology" (i.e., the mean values in the 12 cases selected). However, in these sections, you show that

"anomalous" cases can be a challenge for RAMEFI. Do you expect this issue to be partially or totally solved when RAMEFI is trained on more data? Is this explored in the Part II paper?

> We normalise theta and wind speed to remove location-specific effects, such as exposure near a coast or hillside, i.e., consider local anomalies. However, we do not normalise pressure or theta with the synoptic background state of a given storm, i.e., we do not consider anomalous cyclone developments, as, using irregular distributed station observations, we would not be able to find a mean temperature or core pressure for normalising these parameters anyway as discussed in Section 7.4. For the sake of simplicity and since it seems to be working well for most cyclones, but also ok for anomalous cases, we argue that the spatial independence has more advantages than disadvantages.

> Part II will not focus on a newly trained RF but will use RAMEFI trained on the introduced 12 cases on two data sets of almost 20 years. Including more storms will probably not have a significant effect on the training.

Conclusions:
- Lines 574-575: What surface values do you think could best distinguish CJ from CS (apart from p)? Maybe spatial temperature/theta gradients could be useful? Otherwise, the most obvious to me would be cloud cover, but I'm not sure how many stations would have radiation/sunshine measurements.

> As mentioned in the manuscript and the comment before, normalising the pressure by the core pressure might help distinguish the two features. However, the problems with this outweigh the advantages as discussed. Same for theta. Gradients are not available for station data. Cloud cover might be an interesting parameter indeed but is not available in our data set and is probably difficult to obtain for a data set of 20 years over most of Europe.

- Lines 588-589: This is indeed an encouraging result, but this sentence is probably too bold, given you've just looked at 12 cases in observations vs 1 reanalysis dataset. Maybe you could replace "should" with "could"?

> This has been changed accordingly (line 624).

---

## Referee Report (RR1)

Thank you for carefully taking most of my previous comments into account and revising the manuscript accordingly. The revised manuscript is certainly improved. The reorganisation of the structure and the rephrasing / clearer explanation of the machine learning terminology certainly helped. I have a few minor comments remaining:

- The revisions made to the manuscript in an attempt to address my previous major comment #2 ("Limitations of the method should be clearly highlighted) are quite well hidden in the manuscript and only amount to a few added sentences. I am of the opinion that the limitations should be highlighted in the conclusions section as well.

- Section 2.1 / my previous minor comment #8. In the revised manuscript, in lines 97-98 it is stated that "*the WCB is the main cause for long-lasting precipitation (Catto, 2016). Furthermore, the WCB can be the cause of strong convection along the cold front*". Then in line 102-103 it is written: "*the WJ is usually characterised by positive temperature anomalies, decreasing pressure with time and little or no precipitation.*" I cannot see how these two statements are consistent with each other especially considering in lines 93-94 it is stated that the WJ and the WCB are the same thing: "..*WJ is associated with a warm air flow, typically ahead of and later ascending above the surface cold front, often referred to as the warm conveyor belt (WCB)*" This part of the manuscript needs to be revised.

- Very minor comment: Appendix C is now referred to before Appendix A and B. Consider changing the order of the Appendices.

- Section 3.1 / my previous major comment #1c.1 (how the 12 storms were selected). I still find it a bit unclear. Is it the 12 storms with the largest SSI or is it storms with a non-zero SSI and then a subjective choice to make sure a diverse range of storms is chosen? Please clarify and add a few more details to the manuscript.

- Line 241 "the approach is independent of temporal evolutions beyond 1 h". This is unclear - I think what is meant is that the approach is independent of temporal resolution greater than 1 h / time difference less than 1 h. Please revise.

- Line 243. "in several selected case studies" If this is the 12 case studies it would be clearer to write 12 rather than "several"

- Line 584. "Within the warm sector". Should this read within the warm jet?

- Figure 6 and 7. Although CEP is defined in the text, it would help a reader to add this into the captions. Furthermore, the x-label "Forecast value" does not seem consistent with the revised terminology in the manuscript.

- Figure 8, 9 and 10. I really feel that the delta symbol problem will need to be fixed now or during the copy-editing stage. At worst, an explanation of this symbol needs to be added to the caption.

---

## Author Response (AR2)

**Review of "Objective Identification of high wind features within extratropical cyclones usinga probabilistic random forest (RAMEFI). Part 1: Method and illustrative case studies"**
by L. Eisenstein, B. Schulz, G. A. Qadir, J. G. Pinto and P. Knippertz
10.5194/wcd-2022-29

We would like to thank the reviewers and editor again for their review of our manuscript.
Below we addressed the remaining comments of Reviewer 1 in blue. Text changes are included in italics when suitable. Line numbers correspond to the revised manuscript

Thank you for carefully taking most of my previous comments into account and revising the manuscript accordingly. The revised manuscript is certainly improved. The reorganisation of the structure and the rephrasing / clearer explanation of the machine learning terminology certainly helped. I have a few minor comments remaining:

- The revisions made to the manuscript in an attempt to address my previous major comment #2 ("Limitations of the method should be clearly highlighted) are quite well hidden in the manuscript and only amount to a few added sentences. I am of the opinion that the limitations should be highlighted in the conclusions section as well.
  We added the requirement of hourly resolution again in the conclusions: "*Being independent of spatial behaviour or gradients* and only requiring 1h tendencies*, the approach is very flexible and can be applied to single stations or grid points and various data sets with differing grid spacing. However, due to the fast movement of meteorological features in stormy situations hourly resolution is required, making the algorithm inapplicable to some climate data sets*." (lines 608 – 609)
  Furthermore, the following sentence regarding the wind threshold was added: "*We note that we set a v threshold of 0.8 to focus on high wind areas. However, we do not expect the RF to be sensitive to small changes in the threshold and, in principle, the RF can be applied to wind speeds below this.*" (lines 605 – 607)
  A note on usability outside of the training area was already added in the conclusions section in our last revision (lines 626 – 628).
- Section 2.1 / my previous minor comment #8. In the revised manuscript, in lines 97-98 it is stated that "the WCB is the main cause for long-lasting precipitation (Catto, 2016). Furthermore, the WCB can be the cause of strong convection along the cold front". Then in line 102-103 it is written: "the WJ is usually characterised by positive temperature anomalies, decreasing pressure with time and little or no precipitation." I cannot see how these two statements are consistent with each other especially considering in lines 93-94 it is stated that the WJ and the WCB are the same thing: "..WJ is associated with a warm air flow, typically ahead of and later ascending above the surface cold front, often referred to as the warm conveyor belt (WCB)" This part of the manuscript needs to be revised.
  Again, we are sorry for the confusion. The last statement is meant as the WJ is associated with the **early stages** of a WCB **before** it ascends and causes the discussed precipitation. It is not associated with the whole development of a WCB.
  We added the following sentence for clarification: "*An important feature in extratropical cyclones is the warm conveyor belt (WCB; […]). It starts near the surface ahead of the surface cold front and later ascends above it. […]*
  *Here, we focus on the early stages of the WCB while it is still near the surface and can cause high winds there and refer to it as the WJ.*" (lines 94 – 100)
- Very minor comment: Appendix C is now referred to before Appendix A and B. Consider changing the order of the Appendices.
  This has been changed accordingly.
- Section 3.1 / my previous major comment #1c.1 (how the 12 storms were selected). I still find it a bit unclear. Is it the 12 storms with the largest SSI or is it storms with a non-zero SSI and then a subjective choice to make sure a diverse range of storms is chosen? Please clarify and add a few more details to the manuscript.

It is a combination of both. We included the eight storms with the largest SSI and added 4 more storms with an SSI > 3 based on a subjective choice.

We adjusted the following sentence accordingly: "*This includes the eight  winter storms with the highest SSI during this time period plus four subjectively chosen  more moderate storms to capture a healthy diversity of cyclones and features.*" (lines 202 – 204)

- Line 241 "the approach is independent of temporal evolutions beyond 1 h". This is unclear - I think what is meant is that the approach is independent of temporal resolution greater than 1 h / time difference less than 1 h. Please revise.

  What we wanted to say here was that the approach does not need a full time series of several hours but only the evolution since the last hour (--> tendencies).

  We changed the sentence to: "*The approach evaluates each 1h interval independently.*" (lines 241 – 242)

- Line 243. "in several selected case studies" If this is the 12 case studies it would be clearer to write 12 rather than "several"

  This has been changed accordingly.

- Line 584. "Within the warm sector". Should this read within the warm jet?

  As there is no "Within the warm sector" in line 584, we believe it to be line 501. Indeed, it should read warm jet. This has been changed accordingly.

- Figure 6 and 7. Although CEP is defined in the text, it would help a reader to add this into the captions. Furthermore, the x-label "Forecast value" does not seem consistent with the revised terminology in the manuscript.

  We wrote out the abbreviation CEP in the y-label and changed the x-label according to the terminology used in the manuscript.

- Figure 8, 9 and 10. I really feel that the delta symbol problem will need to be fixed now or during the copy-editing stage. At worst, an explanation of this symbol needs to be added to the caption.

  The Delta symbol seems to be displayed inaccurately/as some other symbol in some older versions of PDF readers or when printed. However, we could not reproduce this problem using various PDF readers ourselves, and we hope it is an isolated incident, such that we did not add an explanation to the caption.